# Online Learning and Inference for Cox Proportional Hazards Models Using Renewable Sieve Estimation

Mengtong Hu [1]   Ziyang Gong [2]   Xu Shi [1]   Ling Zhou [2]   Peter X.K. Song [1]

## Abstract

Online learning for the Cox model is challenging because its partial likelihood is non-decomposable, with each risk set requiring a summation over all eligible samples. We propose Collaborative Operation of Linked Survival Analysis (COLSA), an online learning framework that replaces the partial likelihood with the full likelihood using sieve approximation of the baseline hazard. The proposed likelihood function is decomposable and eliminates the need to store historical data in memory, enabling efficient online updates. Moreover, COLSA maintains sufficient statistics for a higher-order basis and employs data-driven basis projection to adaptively scale model complexity to the effective sample size. Unlike existing online Cox methods, COLSA achieves asymptotic normality and attains the same statistical efficiency as the pooled-data partial likelihood estimator, without accessing full data and only requiring constant memory. Simulation studies and application to kidney transplant data demonstrate that COLSA outperforms existing online methods and matches the performance of full-data estimation.

## 1. Introduction

Real-time decision-making, ranging from patient mortality risk evaluation in healthcare to the prevention of catastrophic failures in industrial monitoring systems, increasingly depends on the analysis of continuous streams of time-to-event data. Although online learning has been extensively studied in regression and classification settings, adapting these efficient frameworks to achieve rigorously accurate statistical survival analysis remains a fundamental challenge. The Cox proportional hazards model, the gold standard for survival analysis, relies on a partial likelihood function constructed from the relative ordering of event times across the entire dataset (Cox, 1972). This global ranking requirement introduces a bottleneck, as calculating the loss function typically requires access to the full historical dataset to construct global risk sets.

This dependency is not merely a computational inconvenience but a critical barrier to data privacy and scalability. A closely related bottleneck arises in Federated Learning, where data cannot be centralized due to privacy constraints. In this setting, standard divide-and-conquer approaches often circumvent this issue by transmitting raw event times to a central server to reconstruct risk sets, or by performing high-frequency iterative communication to align local gradients (Lu et al., 2015; 2021; Duan et al., 2020; Li et al., 2022; Imakura et al., 2023). Both strategies undermine data privacy and incur significant storage and communication overhead (Brink et al., 2022; Andreux et al., 2020). Existing online survival methods (Wu et al., 2021) instead approximate the baseline hazard using fixed-dimensional piecewise-constant functions or employ stochastic gradient descent (SGD) with mini-batch approximations (Tarkhan & Simon, 2024; Zeng et al., 2026). While computationally efficient, these approximations can degrade statistical efficiency, introducing potential bias or variance inflation that is problematic for high-stakes statistical inference.

In this work, we propose COLSA (Collaborative Operation of Linked Survival Analysis), a novel online learning framework that overcomes the global risk set problem without retaining historical data. We approach the online survival analysis problem through the lens of Renewable Sieve Estimation. Instead of a fixed parametric form, we approximate the infinite-dimensional log-baseline hazard using Bernstein basis polynomials.

COLSA enables one-shot model updates by adapting the renewable estimating equation framework introduced by Luo & Song (2020) for fixed parameter dimensions. When a new data batch arrives, the algorithm updates the global estimator using only the summary statistics from the previous batches, eliminating the need for risk-set construction

[1]Department of Biostatistics, University of Michigan, Ann Arbor, USA [2]Center of Statistical Research and School of Statistics and Data Science, Southwestern University of Finance and Economics, Chengdu, China. Correspondence to: Peter Song <pxsong@umich.edu>.

*Proceedings of the 43rd International Conference on Machine Learning*, Seoul, South Korea. PMLR 306, 2026. Copyright 2026 by the author(s).

or iterative communication. We theoretically prove that the renewable estimator is consistent and asymptotically normal, achieving efficiency equivalent to that of a centralized oracle estimator with access to the entire data stream.

In addition to the algorithm and theory of renewable estimation for Cox Models, one of our primary contributions is the dynamic pre-estimation strategy that allows the model complexity, i.e., polynomial degree $p$, to grow adaptively as the data size grows (Quan & Lin, 2024). In standard online learning, the parameter space is usually fixed. In contrast, COLSA introduces a projection mechanism that supports dynamic approximation with increasing polynomial degrees. This guarantees that the approximation error decreases asymptotically with the data stream, achieving the optimal convergence at the rate of the growing total sample size.

We validate COLSA's performance through extensive simulations, demonstrating its asymptotic equivalence to the oracle estimator. Furthermore, we apply COLSA to the kideny transplation related data from large-scale Scientific Registry of Transplant Recipients (SRTR) database (Rao et al., 2007) and cancer gene expression data from The Cancer Genome Atlas (TCGA) (Colaprico et al., 2016), where it successfully replicates centralized findings and identifies significant risk factors missed by relevant online survival methods.

## 2. Related Work

**Distributed and Federated Survival Analysis.** Data privacy regulations in healthcare have driven the development of distributed algorithms for the Cox model. Early approaches like WebDISCO (Lu et al., 2015) and ODAC (Luo et al., 2022) achieve exact inference by reconstructing global risk sets through the exchange of event times across sites or batches. However, these methods typically require iterative communications or explicit sharing of risk-set information, leading to poor scalability under network latency and potential violations of privacy constraints. Moreover, these approaches are infeasible in online settings, where data batches arrive sequentially and historical raw data may become unavailable. While modern Federated Learning (FL) frameworks like FedAvg (McMahan et al., 2017) reduce communication frequency, they primarily function as optimization algorithms for machine learning models rather than principled frameworks for rigorous statistical inference. In addition, they are not naturally suited to the non-decomposable structure of the Cox partial likelihood and often rely on iterative communication rounds to approximate global risk sets.

**Online Learning for Survival Outcomes.** Online learning for survival analysis is constrained by both memory lim-

itations and computational cost. Existing online survival methods often employ SGD on approximating global partial likelihood. For example, Tarkhan & Simon (2024) proposes an approximation of the non-decomposable risk set denominator using scaled mini-batch sums, thereby enabling SGD-based optimization. However, Zeng et al. (2026) demonstrates that while computationally efficient, mini-batch SGD suffers from variance inflation due to batch noise, leading to sub-optimal statistical efficiency. To minimize bias, empirical evidence suggests the optimization process requires over 100 epochs, which is infeasible in streaming data. Moreover, first-order optimization methods typically do not retain the Hessian information, which is required for valid multivariate inference, such as confidence interval construction and hypothesis testing. In contrast, COLSA employs a second-order updating scheme that preserves the full covariance structure, ensuring both scalability and oracle semiparametric efficiency.

Wu et al. (2021) proposes to use cumulatively updated estimating equations (CUEE) to estimate the baseline hazard, approximated through a piecewise constant function, hereinafter referred to as the online survival method. For an accurate approximation of the cumulative baseline hazard using the piecewise-constant approach, a sufficiently fine partition of the event time is necessary. The method assumes large sample sizes within each local batch to guarantee consistency of the estimated hazard ratios. In contrast, COLSA leverages the global smoothness of Bernstein polynomials. By decoupling the dimension expansion from the gradient update, COLSA achieves faster convergence rates for smooth hazards and remains robust even when batch sizes are small. From a theoretical standpoint, the estimation consistency of the CUEE estimator requires a stringent regularity condition, and may fail to hold where streaming data sets $K \to \infty$ (Luo & Song, 2020).

**Dynamic Sieve Estimation.** Sieve estimators, which approximate infinite-dimensional parameters using basis functions that grow with sample size, are well-established in offline statistical inference literature (Shen & Wong, 1994). In survival analysis, polynomial-based Cox models are commonly estimated using sieve maximum likelihood estimation (MLE) (Shen, 1998; Shen & Wong, 1994). A large number of existing studies in the broader survival literature have adopted polynomial splines and sieve MLE to a variety of settings, such as interval-censored data (Zhou et al., 2017; Huang & Rossini, 1997; Zhang et al., 2010), panel count data (Wellner & Zhang, 2007; Lu et al., 2009), and bundled parameters estimation (Ding & Nan, 2011; Zhao et al., 2017; Tang et al., 2023). Unique to streaming data, the cumulative data size increases with the arrival of new batches, so that it becomes necessary to increase the polynomial degree to control approximation error dynamically with increased smoothness. Updating estimates when the current

model is defined on a lower-dimensional basis poses a non-trivial challenge. Building on the pre-estimation strategy of (Quan & Lin, 2024) for one-pass nonparametric estimation in streaming data, we introduce preparatory intermediate statistics along with transformation matrices that enable dynamic approximation with increasing Bernstein polynomial degrees. Quan & Lin (2024) focus mainly on point estimation but not the critical issue of statistical inference. Our work presents a novel projection mechanism based on the recursive structure of Bernstein polynomials, effectively bridging the gap between classical semi-parametric theory and modern online learning algorithms.

## 3. Methodology

We consider a streaming data setting where data batches arrive sequentially indexed by $k = 1, \cdots, K$, where $K$ denotes the total number of batches. The $k$-th batch contains $n_k$ independent and identically (i.i.d.) samples, contributing to a total sample size $N_k = \sum_{j=1}^{k} n_j$.

### 3.1. Problem Formulation

For the $i$-th individual from the $k$-th batch, let $T_{ki}$ be the event time and $C_{ki}$ be the censoring time. We observe the triplet $\mathcal{O}_{ki} = (X_{ki}, Y_{ki}, \Delta_{ki})$, where $X_{ki} \in \mathbb{R}^d$ is the covariate vector, $Y_{ki} = \min(T_{ki}, C_{ki})$ is the observed time, and $\Delta_{ki} = \mathbb{I}(T_{ki} \leq C_{ki})$ is the event indicator. We assume non-informative censoring. The Cox proportional hazards model defines the hazard function at time $t$ given covariates $X$ as:

$$\lambda(t|X) = \lambda_0(t) \exp(X^\top \beta), \qquad (1)$$

where $\beta \in \mathbb{R}^d$ is the regression coefficient vector and $\lambda_0(t)$ is the unspecified baseline hazard function. We use $\tau$ to denote truncation time which is the maximum follow-up time.

**The Non-Decomposable Loss Challenge:** Standard MLE for the Cox model maximizes the partial likelihood. The log-partial likelihood contribution from the $k$-th batch can be written as,

$$\sum_{i=1}^{n_k} \Delta_{ki} \ln \left\{ \frac{\exp(X_{ki}^\top \beta)}{\sum_{k'=1}^{K} \sum_{j=1}^{n_{k'}} \mathbb{I}(Y_{k'j} \geq Y_{ki}) \exp(X_{k'j}^\top \beta)} \right\}$$

The log-partial likelihood term for subject $i$ requires summing over the risk set $\{j : Y_j \geq Y_i\}$. In a streaming or federated setting, constructing the risk set requires accessing raw data from all previous batches, which compromises privacy constraints and exceeds memory limits.

### 3.2. Sieve Approximation via Bernstein Polynomials

To eliminate global risk sets from estimation, we approximate the true log-baseline hazard $g_0(t) = \log \lambda_0(t)$ using

the sieve Maximum Likelihood Estimation method (Shen & Wong, 1994). The individual log-likelihood function is given by

$$\Delta_{ki}\{X_{ki}^\top \beta + g(Y_{ki})\} - \left[ \int_0^{Y_{ki}} \exp\{g(s)\}ds \right] \exp(X_{ki}^\top \beta).$$

Let $g(t)$ be defined on the support $[0, \tau]$, where $\tau < \infty$ is the maximum follow-up time, also known as truncation time and let $\mathcal{S}$ denote the collection of bounded and continuous functions $g$. We consider the estimation of $\theta = (\beta, g)$ defined in the parameter space of $\Theta = \mathbb{R}^d \times \mathcal{S}$, where the true parameter $\theta_0 = (\beta_0, g_0)$. We select **Bernstein basis polynomials** for estimating $g$ for their optimal shape-preserving properties and numerical stability, as well as their recursive structure (Peña, 1997; Farouki & Goodman, 1996), which enables the dynamic expansion of the parameter space (see Sec 3.4).

We approximate $g(t)$ using a polynomial of degree $p$:

$$g^{(p)}(t; \gamma) = \sum_{j=0}^{p} \gamma_j B_j(t, p),$$

where $\gamma = (\gamma_0, \ldots, \gamma_p)^\top$ are the coefficients, and the Bernstein basis functions are defined as:

$$B_j(t, p) = \binom{p}{j} \left( \frac{t}{\tau} \right)^j \left( 1 - \frac{t}{\tau} \right)^{p-j}.$$

This transforms the infinite-dimensional parameter estimation into a finite-dimensional problem over the parameter space $\mathbb{R}^{d+p+1}$.

The separable log-likelihood contribution for the $k$-th batch, independent of global risk sets, becomes:

$$\ell_k(\beta, \gamma) = \sum_{i=1}^{n_k} \Big[ \Delta_{ki}\{X_{ki}^\top \beta + g^{(p)}(Y_{ki}; \gamma)\}$$
$$- \int_0^{Y_{ki}} \exp\{g^{(p)}(s; \gamma)\}ds \cdot \exp(X_{ki}^\top \beta) \Big].$$

In the subsequent algorithm descriptions in Section 3.4, with a slight abuse of notation, let $\theta^{(p)}$ denote the finite-dimensional coefficient vector $(\beta, \gamma) \in \mathbb{R}^{d+p+1}$, and denote the global likelihood $\ell(\theta) = \sum_{k=1}^{K} \ell_k(\theta)$.

### 3.3. Online Renewable Estimation

To update the model online without storing historical data, we employ a **renewable estimation** framework (Luo & Song, 2020). This is a second-order method that approximates the historical likelihood using a quadratic expansion around the previous best estimate.

Let $\tilde{\theta}_{k-1}$ be the estimator derived from previous $k - 1$ batches. When $k$-th batch arrives, we compute the local

gradients $U_k(\theta) = \nabla\ell_k(\theta)$ and negative Hessian $H_k(\theta) = -\nabla^2\ell_k(\theta)$ based *only* on the current batch data.

The global online estimator $\tilde{\theta}_k$ is obtained by solving the **incremental score equation**:

$$U_k(\tilde{\theta}_k) + \sum_{j=1}^{k-1} H_j(\tilde{\theta}_j)(\tilde{\theta}_{k-1} - \tilde{\theta}_k) = 0. \tag{2}$$

For ease of presentation, we define $\tilde{H}_{k-1} := \sum_{j=1}^{k-1} H_j(\tilde{\theta}_j)$ which is the cumulative negative Hessian representing the curvature of the historical loss. This formulation effectively combines the new data's gradient with a historical prior derived from the accumulated Hessian matrices.

## 3.4. Dynamic Implementation with Growing Sieve Size

A critical limitation of standard online learning algorithms is the assumption of a fixed parameter space dimension. To ensure the consistency of the sieve estimator, the polynomial degree $p$ should grow with the sample size ($p \propto N^\nu$, for some $\nu \in (0, 1)$). However, increasing the model complexity in an online setting is challenging: the historical summary statistics reside in a lower-dimensional space and cannot be directly combined with new data. Motivated by the pre-estimation strategy proposed by (Quan & Lin, 2024) to dynamically estimate the coefficients of an expanded set of orthonormal basis functions in streaming data, we propose a two-stage strategy: **Progressive Activation stage** that updates the active model and **Pre-estimation stage** that maintains a higher-dimensional "shadow" sufficient statistic for future smooth transitions.

### 3.4.1. DEGREE INCREASE VIA BERNSTEIN BASIS EXPANSION

We leverage the recursive structure of Bernstein polynomials, which allows a polynomial of degree $p$ to be exactly represented in a higher degree $p'$ space without loss of information, such that

$$B(t, p) = R_{p \to p'}^\top B(t, p'). \tag{3}$$

This is governed by the **degree elevation matrix** $R_{p \to p'}$, defined recursively as:

$$R_{p \to p'} = R_{p'-1 \to p'} \times \cdots \times R_{p \to p+1},$$

where each one-step elevation matrix $R_{p \to p+1} \in \mathbb{R}^{(p+2)\times(p+1)}$ has entries:

$$R_{p \to p+1}[i, i] = \frac{p+1-i}{p+1}, \quad R_{p \to p+1}[i+1, i] = \frac{i+1}{p+1},$$

for $i = 0, \ldots, p$. This ensures that for any coefficient vector $\gamma^{(p)}$, there exists a higher-degree equivalent $\gamma^{(p')} = R_{p \to p'}\gamma^{(p)}$.

### 3.4.2. THE TWO-STAGE UPDATE MECHANISM

Let $\tilde{\theta}_{k-1}^{(p_{k-1})}$ denote the estimator from the previous batch using active degree $p_{k-1}$. Crucially, we assume the algorithm has also maintained a **shadow** negative Hessian matrix $\tilde{H}_{k-1}^{(\bar{p}_{k-1})}$ at a "look-ahead" degree $\bar{p}_{k-1}$, where $\bar{p}_{k-1} > p_{k-1}$.

When the $k$-th new data batch arrives, we determine the new required active degree $p_k$ (where $p_{k-1} < p_k < \bar{p}_{k-1}$). The update proceeds in two phases:

**1. Progressive Activation (Active Model Update):** To initialize the optimization at the new degree $p_k$, we project the historical information into the current space:

- **Coefficient Transformation:** The historical coefficients are mapped exactly to the new degree:

$$\tilde{\gamma}_{k-1}^{(p_k)} = R_{p_{k-1} \to p_k}\tilde{\gamma}_{k-1}^{(p_{k-1})}.$$

- **Hessian Projection:** We derive the relevant curvature information for degree $p_k$ from the larger shadow matrix $\tilde{H}^{(\bar{p}_{k-1})}$ via the transformation: Similarly, the projected negative Hessian matrix becomes

$$\tilde{H}_{k-1,\gamma\gamma}^{(p_k)} = R_{p_k \to \bar{p}_{k-1}}^\top \tilde{H}_{k-1,\gamma\gamma}^{(\bar{p}_{k-1})} R_{p_k \to \bar{p}_{k-1}},$$
$$\tilde{H}_{k-1,\gamma\beta}^{(p_k)} = R_{p_k \to \bar{p}_{k-1}}^\top \tilde{H}_{k-1,\gamma\beta}^{(\bar{p}_{k-1})},$$
$$\text{and } \tilde{H}_{k-1,\beta\beta}^{(p_k)} = \tilde{H}_{k-1,\beta\beta}^{(\bar{p}_{k-1})}.$$

With these warm-start values, we obtain the new estimator $\tilde{\theta}_k^{(p_k)}$ by solving the standard incremental score equation

$$U_k\left(\tilde{\theta}_k^{(p_k)}\right) + \tilde{H}_{k-1}^{(p_k)}\left(\tilde{\theta}_{k-1}^{(p_k)} - \tilde{\theta}_k^{(p_k)}\right) = 0. \tag{4}$$

(derived from Eq. 2).

**2. Pre-estimation (Shadow Model Update):** To prepare for future batches, we must update the shadow statistics to an even higher degree $\bar{p}_k$ ($\bar{p}_k > \bar{p}_{k-1} > p_k$) in order to achieve a more accurate approximation of the log-baseline hazard function as the sample sizes accumulate, that is, we must maintain an up-to-date negative Hessian matrix $\tilde{H}_k^{(\bar{p}_k)}$. This pre-estimation step enables smooth and stable future updates.

To achieve this, we need to project the existing negative Hessian matrix $\tilde{H}_{k-1}^{(\bar{p}_{k-1})}$ onto the parameter space corresponding to degree $\bar{p}_k$. Specifically, we first compute

$$\tilde{H}_{k-1,\gamma\gamma}^{(\bar{p}_k)} = R_{\bar{p}_{k-1}\to\bar{p}_k}^{\top\dagger} \tilde{H}_{k-1,\gamma\gamma}^{(\bar{p}_{k-1})} R_{\bar{p}_{k-1}\to\bar{p}_k}^\dagger,$$
$$\tilde{H}_{k-1,\gamma\beta}^{(\bar{p}_k)} = \left(\tilde{H}_{k-1,\beta\gamma}^{(\bar{p}_k)}\right)^\top = R_{\bar{p}_{k-1}\to\bar{p}_k}^{\top\dagger} \tilde{H}_{k-1,\gamma\beta}^{(\bar{p}_{k-1})},$$

and $\tilde{H}_{k-1,\beta\beta}^{(\bar{p}_k)} = \tilde{H}_{k-1,\beta\beta}^{(\bar{p}_{k-1})}$, where

$$R_{\bar{p}_{k-1}\to\bar{p}_k}^\dagger := (R_{\bar{p}_{k-1}\to\bar{p}_k}^\top R_{\bar{p}_{k-1}\to\bar{p}_k})^{-1} R_{\bar{p}_{k-1}\to\bar{p}_k}^\top$$

is the Moore–Penrose pseudoinverse and $R_{\bar{p}_{k-1}\to\bar{p}_k}^{\top\dagger} = (R_{\bar{p}_{k-1}\to\bar{p}_k}^{\dagger})^{\top}$. We then update the negative Hessian matrix with newly arriving data at site $k$ using the elevated coefficients $\tilde{\theta}_k^{(\bar{p}_k)} = \left(\tilde{\beta}_k, R_{p_k\to\bar{p}_k}\tilde{\gamma}_k^{(p_k)}\right)$ via

$$\tilde{H}_k^{(\bar{p}_k)} = \tilde{H}_{k-1}^{(\bar{p}_k)} + H_k\left(\tilde{\theta}_k^{(\bar{p}_k)}\right).$$

*Remark* 3.1 (Least-Squares Interpretation). The degree elevation relation between Bernstein bases (see equation (3)) defines a linear embedding from lower-degree to higher-degree spaces. Since the transformation matrix $R_{p\to p'}$ is generally non-square, its inverse is not well-defined. We therefore use the Moore–Penrose which yields the best reverse mapping in the least-squares sense. This ensures stable and consistent elevation of negative Hessian matrices polynomial degrees.

This look-ahead strategy ensures that when the active degree $p$ eventually grows, the historical Hessian information is already available in the desired high-dimensional space, preventing information loss and optimization instability. We refer to this projected historical Hessian as the "Working Hessian" in contrast to the ideal "Oracle Hessian" which would have been obtained if the future dimension were known a priori. Theorem D.4 in Appendix D.1 shows the relative error is $O(\rho^{-1/\nu}) + O(p_k^{1/2-q})$: the first term is controlled by choosing $\rho$ sufficiently large, while the second vanishes as $p_k \to \infty$.

The overall implementation of COLSA can be found in Figure 1 and Algorithm 1 in Appendix.

### 3.5. Hyperparameter Selection Strategy

The performance of COLSA relies on three key hyperparameters that govern the sieve growth rate: the scaling constant $C$, the growth rate $\nu$, and the pre-estimation expansion factor $\rho$. We adopt a fully data-driven strategy to select these parameters adaptively as the sample size $N_k$ increases. Consequently, the active degree at batch $k$ is determined by $p_k = \lfloor CN_k^{\nu}\rfloor$.

**1. Sieve Growth Rate ($\nu$):** Theorems 4.1 and 4.2 require $1/(2(q+1)) < \nu < 1/(2q)$ for optimal convergence. Under a standard smoothness assumption (e.g., $g_0$ is twice continuously differentiable, $q = 2$), this restricts $\nu$ to $(0.167, 0.250)$. We therefore recommend $\nu = 0.2$ as a theoretically justified default, rather than treating it as a tunable parameter.

**2. Pre-estimation Factor ($\rho$):** Theorems 4.1 and 4.2 only require $\rho > 1$. Theorem D.4 shows the relative Working Hessian error is $O(\rho^{-1/\nu})$. For $\nu = 0.2$, $\rho = 2$ yields error $\lesssim 5\%$ with modest memory overhead; $\rho = 3$ reduces error to $\lesssim 1\%$. The empirical results show negligible variation in performance across all streaming settings ($K$) for

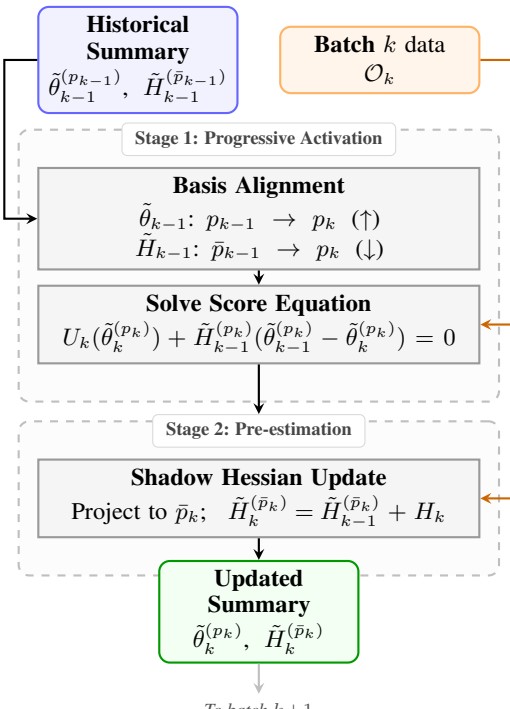

*Figure 1.* Schematic diagram of the COLSA algorithm at Batch $k$. Historical summaries are first projected to the appropriate dimensionality before being updated with the current batch data ($\mathcal{O}_k$). The process is strictly divided into an active model update and a shadow matrix preparation.

varying degrees of $\rho \in \{1.5, 2.0, 2.5, 3.0\}$. Therefore, we recommend $\rho = 2$ as a default.

**3. Scaling Constant ($C$) via AIC:** This is the only parameter that requires data-driven selection. We estimate $C$ using AIC on the first batch by search over $p_1 \in \{3, \ldots, \lfloor n_1^{1/3}\rfloor\}$, then set $C = p_1^*/n_1^{\nu}$. When $n_1$ is small ($< 200$), we recommend a conservative default $p_1 = 3$ or $4$. Simulations show COLSA is robust to moderate variations in $C$.

## 4. Theoretical Guarantees

Let $\theta_0 = (\beta_0, g_0) \in \Theta := \mathbb{R}^d \times L^2[0,\tau]$ denote the true parameter, where $\beta_0$ is the true regression coefficient, $g_0$ is the true log-baseline hazard, and $L^2[0,\tau]$ is the space of square-integrable functions on $[0,\tau]$. Let $\|\cdot\|$ denote the Euclidean norm for vector. Define the distance on the parameter space as

$$d(\theta_1, \theta_2) := \|\beta_1 - \beta_2\| + \|g_1 - g_2\|_{L^2},$$

where $\|g\|_{L^2} := \left(\int_0^\tau g(t)^2\,dt\right)^{1/2}$ is the $L^2$-norm on $[0,\tau]$. In this section, we establish the consistency and asymptotic normality of the COLSA estimator $\tilde{\theta}_k$, showing that it achieves oracle efficiency as the cumulative sample size $N_k = \sum_{j=1}^{k} n_j \to \infty$. We assume $n_k = O(N_{k-1})$, i.e., no

single batch dominates. The sieve dimension $p_k$ grows with $N_k$.

### 4.1. Regularity Conditions

To establish large-sample properties, we require standard regularity conditions for sieve M-estimation (Shen, 1998; Tang et al., 2023):

(C1) The true parameter $\beta_0$ lies within the interior of a compact set $\mathcal{B} \subset \mathbb{R}^d$.

(C2) Let $\mathcal{S}^q([a, b])$ be the collection of bounded functions $f$ on $[a, b]$ with bounded derivatives $f^{(j)}$, $j = 1, \ldots, k$, where the $k$-th derivative $f^{(k)}$ satisfies the $m$-Hölder continuity condition:

$$|f^{(k)}(s) - f^{(k)}(t)| \leq L|s - t|^m \quad \forall s, t \in [a, b],$$

where $k$ is a positive integer and $m \in (0, 1]$ with $q = m + k$, and $L < \infty$ is a constant. The true log-baseline hazard function $g_0(\cdot)$ belongs to $\mathcal{S}^q([0, \tau])$ with $q \geq 2$.

(C3) The covariate $X$ belongs to a bounded subset $\mathcal{X} \subset \mathbb{R}^d$, with density lower-bounded by a constant $c > 0$. Furthermore, $E(XX^\top)$ is nonsingular.

(C4) There exists a truncation time $\tau < \infty$ such that, for some positive constant $\delta_0$, $\Pr(Y > \tau | X) \geq \delta_0$ almost surely with respect to the probability measure of $X$.

(C5) Let $R(t) = \int_0^t \exp(g_0(s))ds$, $V = \beta_0^\top X$, and $U = e^V R(Y)$. There exists $\eta \in (0, 1)$ such that for all $u \in \mathbb{R}^d$ with $\|u\| = 1$,

$$u^\top \text{Var}(X|U, V)u \geq \eta \, u^\top E(XX^\top | U, V)u \quad \text{a.s..}$$

(C6) The information matrix $I(\beta_0)$ is nonsingular, where

$$I(\beta_0) = \int_0^\infty E\left[\{-X + \mu(t)\}^{\otimes 2} \mathbf{1}(U \geq t)\right] dt,$$

with $\mu(t) = E[\mathbf{1}(U \geq t)e^{\beta_0^\top X}X]/E[\mathbf{1}(U \geq t)e^{\beta_0^\top X}]$ and $U$ defined in (C5).

Conditions (C1) and (C2) are standard regularity assumptions in the survival analysis literature (Huang, 1996). Condition (C2) requires sufficient smoothness of the true log-baseline hazard; the parameter $q$ controls the sieve approximation rate. Conditions (C3) and (C4) are standard data quality requirements, implying that covariates exhibit sufficient variation and the censoring rate is not too high. Condition (C5) and (C6) are non-degeneracy assumptions that guarantee identifiability and stable estimation. Condition (C5), proposed by (Wellner & Zhang, 2007), can be verified in many applications when (C3) holds. Condition (C6) is a natural assumption requiring the information matrix to be invertible.

### 4.2. Main Results

We show that despite the non-iterative, one-pass nature of our update, COLSA recovers the statistical properties of the centralized oracle estimator.

**Theorem 4.1** (Consistency). *Under conditions (C1)–(C6) with $\nu < 1/(2q)$ and $\rho > 1$ sufficiently large:*

$$d(\tilde{\theta}_k, \theta_0) = O_p(N_k^{-\min(q\nu, (1-\nu)/2)}).$$

**Theorem 4.2** (Asymptotic Normality & Oracle Efficiency). *Under the conditions of Theorem 4.1, if additionally $\nu > 1/(2(q+1))$, the estimator $\tilde{\beta}_k$ satisfies:*

$$\sqrt{N_k}(\tilde{\beta}_k - \beta_0) \xrightarrow{d} \mathcal{N}(0, I^{-1}(\beta_0)), \qquad (5)$$

*where $I(\beta_0)$ is the efficient Fisher information matrix defined in (C6).*

**Remark (Optimal Convergence Rate):** Theorem 4.2 implies that COLSA achieves the **semiparametric efficiency bound**. The asymptotic variance $I^{-1}(\beta_0)$ is identical to that of the oracle estimator which has access to the full historical dataset. Detailed proofs are provided in the Appendix D.

## 5. Experiments

We evaluate COLSA on both simulated streaming data and the large-scale Scientific Registry of Transplant Recipients (SRTR) database ($N = 48,766$). We benchmark COLSA against four primary baselines: (1) **Oracle**, a centralized, pooled-data analysis, (2) **Meta**, a classical statistical approach that combines batch-specific hazard ratio estimates via inverse-variance weighting (Cochran, 1954) (3) **Online**, a streaming approach, utilizing piecewise-constant estimation on baseline hazards (Wu et al., 2021), and (4) **SGD**, a stochastic gradient descent implementation via the R package coxphSGD (Kosinski, 2017). In the simulation studies, we further include **SGD (Offline)**, which, unlike its streaming counterpart, performs iterative optimization for many epochs by assuming persistent access to previously processed batches.

### 5.1. Simulation Studies

We evaluate COLSA on synthetic data streams distributed with the number of batches $K$ varying to be 6, 20, 50, and 200. Data were generated using a mixture Weibull baseline hazard with correlated covariates with varying batch sizes from 500 to 1500 to mimic the SRTR structure. The detailed data generation scheme and hyperparameter settings are provided in the Appendix B. The simulation results below are averaged over 500 replications and run in RStudio on an Apple M1 Pro processor with 16G of RAM, utilizing parallel processing in multiple cores.

**Results:** We assess the estimation performance using Average Absolute Bias (Abias) to measure the total systematic deviation of the coefficients from their true values and Root Mean Squared Error (RMSE) to assess overall accuracy by combining bias and empirical variance. Additionally, we use Average Coverage Probability (AvgCP) to verify the reliability of the confidence intervals and Mean Basis Functions to monitor the adaptive complexity of COLSA. The simulation

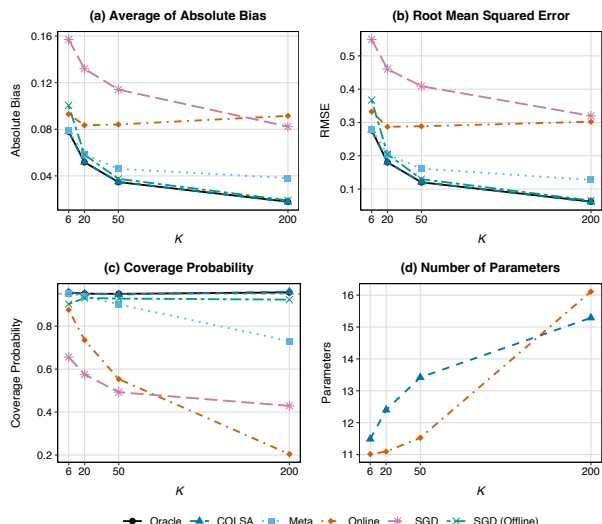

*Figure 2.* Simulation performance comparison across varying numbers of data batches ($K$). (a) Average Absolute Bias (Abias). (b) Root Mean Squared Error (RMSE). (c) Average Coverage Probability (AvgCP). (d) Number of Parameters: Illustrates the adaptive complexity of the COLSA model as the total sample size increases. Methods compared: Oracle (black circle/solid line), COLSA (blue triangles/dashed line), Online (orange diamond/dot-dashed line), SGD (pink star/solid line), and offline SGD ('x' markers/dashed line).

results in Figure 2 demonstrate that COLSA consistently outperforms all other methods, closely matching the performance of the Oracle estimator in both Abias and RMSE. As the number of batches ($K$) increases, COLSA shows a significant reduction in both Abias and RMSE, maintaining a trajectory line nearly identical to the centralized Oracle analysis. Although the multi-epoch SGD variant shows an improved bias reduction with increasing $K$, its convergence remains slower than COLSA and Oracle. This limitation stems from the non-decomposable structure of the Cox partial likelihood: partitioning data into mini-batches restricts each risk set to only batch-local samples, fundamentally deviating from the full-data risk set required for unbiased gradient estimation. Furthermore, COLSA shows strong statistical inference reliability with stable 95% coverage probabilities. Lastly, COLSA adaptively increases its basis functions as more data arrives, reflected in its parameter

growth. The Online method also grows its parameter size, and it eventually exceeds the model size of COLSA.

The Online method is primarily designed for large batches, and the adaptive partitioning strategy fails to construct a sufficiently fine grid approximation when local data is sparse (e.g., $n_k = 100$), leading to significant bias for smooth baseline hazards like the Weibull distribution. To test this limitation, we conduct additional simulations with local sample sizes reduced to $n_k = 100$. In cases where the Online method fails to converge locally, we allow the algorithm to proceed to the next batch to maintain the integrity of the algorithm. As shown in our results, COLSA maintains nearly identical performance to the Oracle even under constrained batch sizes. This is a notable contrast to the Online method, which suffers from two main issues when local data is sparse: computational instability and the lack of a pre-estimation stage. Without that stage, the adaptive basis expansion in the Online method introduces significant bias. In contrast, COLSA's renewable sieve estimator leverages the global smoothness of Bernstein polynomials, allowing it to fit complex hazard functions accurately even with limited data per batch. Full numerical results for these small-batch scenarios are provided in the Appendix C Figure 3. Detailed results for these simulations are provided in the Appendix.

*Table 1.* Storage complexity and computation time.

| Methods | Storage | Time (seconds) | | | |
|---|---|---|---|---|---|
| | | $K=6$ | $K=20$ | $K=50$ | $K=200$ |
| *Streaming (no full data access)* | | | | | |
| SGD | $O(d)$ | 0.062 | 0.178 | 0.386 | 1.606 |
| Meta | $O(Kd^2)$ | 0.033 | 0.108 | 0.284 | 1.585 |
| Online | $O((d+J)^2)$ | 0.202 | 0.461 | 1.096 | 4.516 |
| COLSA | $O((d+\bar{p})^2)$ | 0.734 | 1.565 | 3.806 | 28.36 |
| *Full data access required* | | | | | |
| Oracle | $O(Nd)$ | 0.025 | 0.064 | 0.135 | 0.693 |
| SGD (Offline) | $O(Nd)$ | 2.622 | 6.786 | 14.63 | 58.14 |

*Note:* $N$ = total sample size; $K$ = total number of batches; $d$ = covariate dimension; $\bar{p}$ = pre-estimation degree; $J$ = number of partitions.

Table 1 summarizes the computational trade-offs. Among streaming methods, SGD requires only $O(d)$ storage but exhibits substantial bias as shown in Figure 2. COLSA stores the cumulative Hessian with $O((d + \bar{p})^2)$ complexity, where $\bar{p} = O(N^\nu)$, achieving Oracle-level efficiency without retaining raw data. In terms of computation time, COLSA is slower than Meta and Online per batch due to the sieve basis expansion and numerical integration, but remains substantially faster than multi-epoch SGD (Offline) which requires 58.14s versus 28.36s at $K = 200$. Oracle and SGD (Offline) both require $O(Nd)$ storage for full data retention, making them impractical for streaming settings.

## 5.2. Application: OPTN Kidney Transplant Data

We obtain the relevant data from SRTR collected by the Organ Procurement and Transplantation Network (OPTN). The goal was to identify risk factors for 5-year death-censored graft failure (DCGF) without a central processing center that stores all sensitive patient data. Our results illustrate that COLSA successfully replicates the findings of the centralized Oracle analysis in a streaming data environment, where data batches from $49$ individual states were processed sequentially, and the number of samples per batch ranged from $68$ to $5454$. COLSA successfully identifies associations that are either missed by underestimated Z-statistics $\leq 1.96$ or false discoveries caused by inflated Z-statistics $> 1.96$ based on methods (Gender match, donor obesity, recipient and donor race, HLA mismatch, which are underlined in Table 2). The comparison methods struggle to reach statistical inference consistency for sparse categorical variables. The only variable for which COLSA yields different statistical inference results compared to Oracle is the Hispanic recipient's lower risk for DCGF ($1.6\%$ recipient develop DCGF are Hispanic). Given that there is a known disparity in kidney transplant access among different racial demographics, the significant results should be interpreted with caution.

*Table 2.* Hazard Ratios and Z-statistics (in parentheses) for the SRTR Database Analysis.

| Variable | Oracle | COLSA | Meta | Online | SGD |
|---|---|---|---|---|---|
| Recipient Age | 0.98 (-21.30) | 0.98 (-21.23) | 0.98 (-21.09) | 0.98 (-19.44) | 0.98 (-13.74) |
| Donor Age | 1.02 (17.16) | 1.01 (16.78) | 1.01 (16.08) | 1.02 (18.22) | 1.01 (6.93) |
| *Gender Match (Ref: F-F)* | | | | | |
| F-M | 1.08 (2.06) | 1.07 (1.97) | 1.07 (1.95) | 1.13 (3.20) | 1.02 (0.44) |
| M-M | 0.89 (-3.27) | 0.89 (-3.22) | 0.90 (-3.01) | 0.98 (-0.40) | 0.98 (-0.40) |
| M-F | 1.01 (0.32) | 1.01 (0.35) | 0.96 (-1.00) | 1.09 (2.17) | 1.39 (5.79) |
| Recipient Obesity | 1.16 (5.48) | 1.16 (5.37) | 1.18 (5.97) | 1.18 (5.43) | 1.37 (7.05) |
| Donor Obesity | 1.05 (1.63) | 1.05 (1.74) | 1.06 (2.10) | 1.05 (1.72) | 1.53 (9.58) |
| *Recipient Race (Ref: White)* | | | | | |
| Black | 1.52 (13.87) | 1.50 (13.62) | 1.52 (13.46) | 1.52 (12.68) | 1.88 (12.95) |
| Hispanic | 0.91 (-2.22) | 0.93 (-1.58) | 1.02 (0.37) | 0.93 (-1.45) | 1.01 (0.22) |
| Other | 0.80 (-3.62) | 0.82 (-3.16) | 0.88 (-1.96) | 0.82 (-3.06) | 0.91 (-1.12) |
| *Donor Race (Ref: White)* | | | | | |
| Black | 1.30 (7.31) | 1.28 (7.17) | 1.30 (7.38) | 1.31 (6.94) | 1.52 (7.61) |
| Hispanic | 1.04 (0.92) | 1.05 (1.24) | 1.03 (0.64) | 1.09 (1.95) | 0.91 (-1.45) |
| Other | 1.07 (0.88) | 1.08 (0.98) | 1.19 (2.30) | 1.12 (1.42) | 0.96 (-0.35) |
| *HLA Mismatch (Ref: 0)* | | | | | |
| 1-2 MM / 0 DR | 1.26 (2.99) | 1.24 (2.91) | 1.26 (3.04) | 1.58 (6.07) | 1.25 (1.99) |
| 1-2 MM / 1-2 DR | 1.51 (7.32) | 1.48 (7.33) | 1.46 (6.58) | 1.86 (11.96) | 1.09 (1.17) |
| 3-4 MM / 0 DR | 1.34 (3.92) | 1.32 (3.74) | 1.40 (4.38) | 1.67 (6.58) | 0.81 (-1.88) |
| 3-4 MM / 1-2 DR | 1.68 (9.91) | 1.64 (9.95) | 1.61 (8.91) | 2.03 (14.30) | 1.12 (1.65) |
| Donor Deceased | 1.78 (18.67) | 1.74 (18.03) | 1.76 (18.06) | 1.84 (18.42) | 1.37 (6.51) |
| Recipient Diabetes | 0.97 (-0.99) | 0.95 (-1.57) | 0.97 (-1.07) | 0.96 (-1.14) | 1.01 (0.17) |
| Recipient Hypertension | 1.04 (1.29) | 1.03 (0.95) | 1.02 (0.71) | 1.10 (2.75) | 1.86 (13.16) |

## 5.3. Application: TCGN genomics Data

We analyzed RNA-Seq data from The Cancer Genome Atlas (TCGA) for $N = 7,315$ patients across 18 cancer types (each with $\geq 50$ death events), obtained from the R package TCGAbiolinks (Colaprico et al., 2016; Silva et al., 2016; Mounir et al., 2019). Expression counts were normalized via DESeq2 (Love et al., 2014) and $\log_2$-transformed. To establish a ground-truth set of covaraites for testing the estimation

*Table 3.* Hazard Ratios and $Z$-statistics (in parentheses) for the TCGA Pan-Cancer Analysis.

| Variable | Oracle | COLSA | Meta | Online | SGD |
|---|---|---|---|---|---|
| FLNC | 1.06 (4.75) | 1.06 (4.51) | 1.06 (4.69) | 1.19 (9.64) | 1.07 (5.27) |
| GATA4 | 1.03 (5.27) | 1.03 (5.12) | 1.02 (3.01) | 1.07 (7.73) | 1.03 (3.46) |
| PIK3R3 | 0.97 (-1.63) | 0.97 (-1.21) | 0.98 (-0.65) | **0.80 (-7.61)** | **0.95 (-2.06)** |
| ISL2 | 1.03 (2.34) | 1.02 (2.02) | 1.03 (2.62) | 1.32 (15.00) | 1.01 (0.85) |
| TREML3P | 1.05 (4.61) | 1.05 (4.51) | 1.05 (4.16) | 1.02 (1.42) | 1.05 (3.82) |
| SERPINE1 | 1.08 (6.50) | 1.08 (6.20) | 1.05 (3.69) | 0.99 (-0.76) | 1.06 (4.37) |
| PTX3 | 1.05 (5.12) | 1.05 (5.10) | 1.05 (3.91) | **0.93 (-4.47)** | 1.06 (4.78) |
| IL1RAP | 1.00 (0.07) | 0.99 (-0.32) | 0.99 (-0.55) | **1.31 (12.37)** | 0.97 (-1.51) |
| PPP1R1C | 1.04 (4.16) | 1.04 (3.99) | 1.02 (1.74) | 1.18 (10.36) | 1.02 (1.60) |
| CDK6.AS1 | 1.03 (2.31) | 1.04 (2.57) | 1.03 (2.31) | 0.87* (-6.28) | 0.99 (-0.38) |
| CYP19A1 | 1.06 (4.57) | 1.06 (4.59) | 1.05 (3.74) | 1.04 (1.76) | 1.06 (4.18) |
| FKBP1C | 1.01 (0.25) | 1.00 (-0.06) | 1.02 (0.70) | **1.64 (13.61)** | 1.12 (3.94) |
| SEMA3G | 0.90 (-5.94) | 0.90 (-5.66) | 0.90 (-5.57) | 0.84 (-6.26) | 0.90 (-5.29) |
| PGR | 0.99 (-1.49) | 0.99 (-1.41) | 0.99 (-1.19) | **1.03 (2.14)** | 1.01 (0.77) |
| PRLR | 0.98 (-2.89) | 0.98 (-2.14) | 1.00 (0.36) | 0.80 (-17.74) | 1.00 (-0.02) |
| PDCD4 | 0.95 (-2.11) | 0.95 (-1.79) | 0.98 (-0.82) | 0.64 (-12.47) | 0.86 (-5.14) |
| HOXA2 | 1.05 (4.92) | 1.05 (4.60) | 1.05 (4.22) | 1.20 (10.72) | 1.06 (4.69) |
| METTL7B | 1.03 (3.32) | 1.03 (2.82) | 1.02 (1.49) | 1.26 (14.81) | 1.01 (1.06) |
| BCL2 | 0.97 (-1.71) | 0.97 (-1.57) | 0.98 (-1.11) | **0.87 (-5.15)** | 0.92 (-3.90) |
| FLNC.AS1 | 1.02 (0.97) | 1.02 (0.87) | 1.02 (0.87) | 0.93 (-1.70) | 1.03 (1.03) |
| SLC11A1 | 0.98 (-0.98) | 0.99 (-0.85) | 0.99 (-0.53) | **1.11 (4.33)** | 0.94 (-3.17) |
| SEC61G | 1.03 (1.27) | 1.04 (1.58) | 1.04 (1.60) | **0.90 (-3.66)** | 0.98 (-0.78) |
| MYRF | 1.04 (4.10) | 1.04 (3.96) | 1.02 (1.18) | 1.35 (17.95) | 1.01 (0.67) |

*Note:* Underline: False Negative; **Bold**: False Positive; *: Sign reversal.

and inference capacities of the algorithms, we first selected all protein-coding genes by full-subject univariate Cox regression, retaining $4,374$ genes at a Bonferroni-adjusted threshold of $p < 10^{-4}$, and then applied LASSO Cox regression with 5-fold cross-validation to obtain $d = 23$ genes as the candidate set. To enable online learning, each cancer type serves as one streaming batch ($K = 18$) with diverse individual sample sizes ranging from BRCA ($n = 1,073$) to MESO ($n = 85$). The relationship between patient survival and these 23 genes was then analyzed through multivariate Cox regression.

The table reports hazard ratios and Wald Z-statistics for each of the gene candidates. The full-data Oracle serves as the benchmark. At the $|Z| > 1.96$ threshold, it identifies 15 of 23 covariates as significant. COLSA demonstrated high consistency with Oracle (Pearson $r = 0.997$ for Z-statistics; mean absolute log-HR difference $0.003$), correctly recovering the inference results on 22 of the 23 gene candidates. The one discordance is PDCD4 ($Z = -1.79$ vs. $-2.11$), which Oracle itself only marginally declares significant. Inspecting PDCD4's batch-level trajectory, its COLSA Z-statistic oscillates between $-1.0$ and $-2.1$ as successive cancer types are incorporated. Per-cancer univariate Cox Z-statistics for PDCD4 range from $-5.68$ (LGG) and $-3.54$ (KIRC) to $+0.58$ (BLCA) and $+0.39$ (STAD). In comparison, Meta missed 5 Oracle-significant genes, while the Online and SGD disagreed on 12 and 10 decisions, respectively.

## 6. Conclusion

We introduced COLSA, an online learning and inference method for the Cox proportional hazards model. By lever-

aging a renewable sieve estimator with dynamic Bernstein polynomials, COLSA solves the non-decomposable loss problem inherent in survival analysis without requiring the storage of historical data or construction of global risk sets. Theoretical analysis and large-scale experiments confirm that it achieves oracle-level accuracy and statistical efficiency. This establishes COLSA not merely as an approximation method, but as a rigorous statistical tool viable for high-stakes federated and streaming environments. To enable COLSA estimation across distributed data, summary statistics can be transmitted across sites as described in (Hu et al., 2024). The COLSA R package is available at https://github.com/CollaborativeInference/COLSA.

Future research will focus on improving the computational efficiency of the renewable sieve update, potentially through the use of parallelized basis function estimation and advanced numerical integration techniques to mitigate the runtime increase observed as the number of sites $K$ grows. The computation speed may also be significantly improved without hurting accuracy if the integration is performed only over unique event times, rather than for every single observation, thereby avoiding redundant calculations. The numerical integration may be avoided entirely if a constrained spline approximation is used to directly estimate the cumulative baseline hazard function. Additionally, the current framework assumes a fixed dimension for the covariates while the sample size grows. Extending the renewable estimation to handle high-dimensional streaming data with regularized online updates is a critical step. For example, Bayle et al. (2025) proposed an iterative LASSO algorithm for the distributed inference in Cox model. The two-stage sieve degree elevation utilized in COLSA could be integrated into existing online debiased frameworks (Luo et al., 2023; Han et al., 2026) to facilitate dynamic estimation within the nuisance parameter space. Although COLSA enhances privacy by transmitting only summary statistics (gradients and Hessians) rather than raw data, it has not yet provided formal privacy guarantees. Future research for private online learning (Xie et al., 2025; Yuan et al., 2026) could incorporate differential privacy (DP) mechanisms by injecting calibrated noise into the second-order summary statistics.

## Acknowledgements

The authors are grateful for the valuable feedback from the anonymous reviewers, which substantially improved the quality of the paper. Ziyang Gong and Ling Zhou were partially supported by National Key R&D Program of China (2022YFA1003702).

## Impact Statement

This paper presents work whose goal is to advance the field of Machine Learning. There are many potential societal consequences of our work, none of which we feel must be specifically highlighted here.

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

## A. Algorithm

---

**Algorithm 1** Progressive Activation and Pre-estimation in COLSA

---

**input** Collaborative data $\{\mathcal{O}_k\}_{k=1}^{K}$; activation constants $C$, $\nu$; pre-estimation constant $\rho > 1$; initial degree $p_0$.

**output** Estimator $\tilde{\theta}_K^{(p_K)}$; cumulative negative Hessian $\tilde{H}_K^{(\bar{p}_K)}$.

1: **Initialization**
2: $p_0 \leftarrow p_0, \bar{p}_0 \leftarrow \lfloor \rho p_0 \rfloor, N_0 \leftarrow 0$
3: Initialize coefficients $\tilde{\theta}_0^{(p_0)}$ and negative Hessian matrix $\tilde{H}_0^{(\bar{p}_0)}$
4: **for** $k = 1, \ldots, K$ **do**
5: $\quad N_k \leftarrow N_{k-1} + n_k$
6: $\quad$ Set activation and pre-estimation degrees:
7: $\quad p_k \leftarrow \lfloor C N_k^{\nu} \rfloor, \quad \bar{p}_k \leftarrow \lfloor \rho C N_k^{\nu} \rfloor$
8: $\quad$ **Progressive Activation Stage**
9: $\quad$ **if** $p_k > p_{k-1}$ **then**
10: $\quad\quad$ Project coefficients:
11: $\quad\quad \tilde{\gamma}_{k-1}^{(p_k)} \leftarrow R_{p_{k-1} \to p_k} \tilde{\gamma}_{k-1}^{(p_{k-1})}$
12: $\quad\quad$ Project negative Hessian matrix:
13: $\quad\quad \tilde{H}_{k-1,\gamma\gamma}^{(p_k)} \leftarrow R_{p_k \to \bar{p}_{k-1}}^{\top} \tilde{H}_{k-1,\gamma\gamma}^{(\bar{p}_{k-1})} R_{p_k \to \bar{p}_{k-1}}$
14: $\quad\quad \tilde{H}_{k-1,\gamma\beta}^{(p_k)} = \left( \tilde{H}_{k-1,\beta\gamma}^{(p_k)} \right)^{\top} \leftarrow R_{p_k \to \bar{p}_{k-1}}^{\top} \tilde{H}_{k-1,\gamma\beta}^{(\bar{p}_{k-1})}$
15: $\quad\quad \tilde{H}_{k-1,\beta\beta}^{(p_k)} \leftarrow \tilde{H}_{k-1,\beta\beta}^{(\bar{p}_{k-1})}$
16: $\quad$ **end if**
17: $\quad$ Solve estimating equation (2) with initial coefficients $\tilde{\theta}_{k-1}^{(p_k)}$ and negative Hessian matrix $\tilde{H}_{k-1}^{(p_k)}$ to obtain $\tilde{\theta}_k^{(p_k)}$ at degree $p_k$
18: $\quad$ **Pre-estimation Stage**
19: $\quad$ **if** $\bar{p}_k > \bar{p}_{k-1}$ **then**
20: $\quad\quad$ Project previous sensitivity matrix:
21: $\quad\quad \tilde{H}_{k-1,\gamma\gamma}^{(\bar{p}_k)} \leftarrow R_{\bar{p}_{k-1} \to \bar{p}_k}^{\top \dagger} \tilde{H}_{k-1,\gamma\gamma}^{(\bar{p}_{k-1})} R_{\bar{p}_{k-1} \to \bar{p}_k}^{\dagger}$
22: $\quad\quad \tilde{H}_{k-1,\gamma\beta}^{(\bar{p}_k)} = \left( \tilde{H}_{k-1,\beta\gamma}^{(\bar{p}_{k-1})} \right)^{\top} \leftarrow R_{\bar{p}_{k-1} \to \bar{p}_k}^{\top \dagger} \tilde{H}_{k-1,\gamma\beta}^{(\bar{p}_{k-1})}$
23: $\quad\quad \tilde{H}_{k-1,\beta\beta}^{(\bar{p}_k)} \leftarrow \tilde{H}_{k-1,\beta\beta}^{(\bar{p}_{k-1})}$
24: $\quad$ **end if**
25: $\quad$ Update the negative Hessian matrix:
26: $\quad\quad \tilde{H}_k^{(\bar{p}_k)} \leftarrow \tilde{H}_{k-1}^{(\bar{p}_k)} + H_k \left( \tilde{\theta}_k^{(\bar{p}_k)} \right)$
27: $\quad\quad$ where $\tilde{\theta}_k^{(\bar{p}_k)} = \left( \tilde{\beta}_k, R_{p_k \to \bar{p}_k} \tilde{\gamma}_k^{(p_k)} \right)$
28: **end for**

---

## B. Simulation setup

We first generate the full data under an assumed proportional hazards model and then split the data into $K$ batches. The first six batches have fixed sample sizes of 1500, 1500, 1500, 500, 500, and 500, and the rest of the $K - 6$ sites have equal sample sizes of 500 or 100 to mimic the real-data application. We assume that the baseline hazard follows the two-component mixture Weibull distribution with a mixture probability of 0.5, a scale parameter vector of $(10, 20)$, and a shape parameter vector of $(3, 5)$. To mimic the covariate distribution in the data example, we generate covariates $X_1$ to $X_4$ with some correlations: two continuous variables $X_1$ and $X_2$ are drawn from a multivariate normal distribution with a mean vector of $(5, 5)$ and a variance-covariance matrix equal to $(10, 3; 3, 2)$ and one binary covariate $X_3$ is drawn from a Bernoulli distribution with a success probability of 0.8. Motivated by the possible distribution of race and ethnicity variables commonly included in survival analysis (see data example for more details), $X_4$ is a four-level categorical variable with a multinomial probability vector of $(0.2, 0.2, 0.3, 0.3)$ or $(0.1, 0.2, 0.4, 0.5)$ for $X_3 = 0$ and $X_3 = 1$ respectively. For convenience, we use $X$ to denote the vector of all covariates, including the one-hot encoding for $X_4$. The true value of the log hazard ratio is $(0.15, -0.15, 0.3, 0.3, 0.3, 0.3)$. The time-to-event outcome $Y$ is generated through the cumulative

hazard inversion method ((Brilleman et al., 2020; Bender et al., 2005)). The censoring time $C$ is independently generated from an exponential distribution with a rate parameter of 6 such that the event rate is controlled to be around $12\%$.

The following hyperparameter settings are used for both simulation studies and real data analysis.

**COLSA.** The pre-estimation factor is $\rho = 2$. The initial degree $p_0$ is selected via AIC on the first batch, with activation constants $C = p_0/n_1^\nu$ and $\nu = 0.2$, where $n_1$ is the first batch size. The integraion is performed using the Gaussian Quadrature.

**Online.** We implement the adaptive partition method of Wu et al. (2021) with initial partition size $J_0 = 5$. Partition breaks are placed at event-count quantiles. At each update, if any interval has event count $d_j > r \cdot \bar{d}$ (where $\bar{d}$ is the average and $r = 2$), the rightmost such interval is split at its median event time.

**SGD.** We use learning rate $\eta_k = (c\sqrt{k})^{-1}$ with $c = \max(100, 500/\sqrt{K})$, where $k$ is the batch index.

**SGD (Offline).** Same learning rate as SGD, trained for 100 epochs. This method is evaluated only in simulations as it requires retaining all historical data.

## C. Additional simulation results

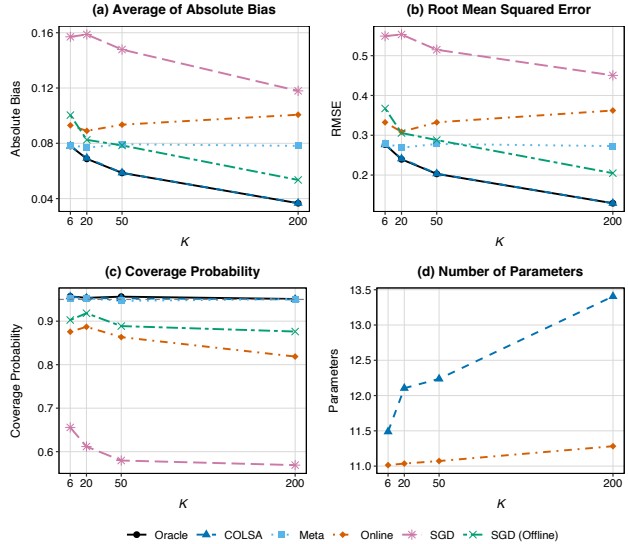

*Figure 3.* Simulation performance comparison across varying numbers of data batches ($K$) with local sample sizes as small as 100. (a) Average Absolute Bias (Abias). (b) Root Mean Squared Error (RMSE). (c) Average Coverage Probability (AvgCP). (d) Number of Parameters: Illustrates the adaptive complexity of the COLSA model as the total sample size increases. Methods compared: Oracle (black circle/solid line), COLSA (blue triangles/dashed line), Online (orange diamond/dot-dashed line), SGD (pink star/solid line), and offline SGD (cross/dashed line).

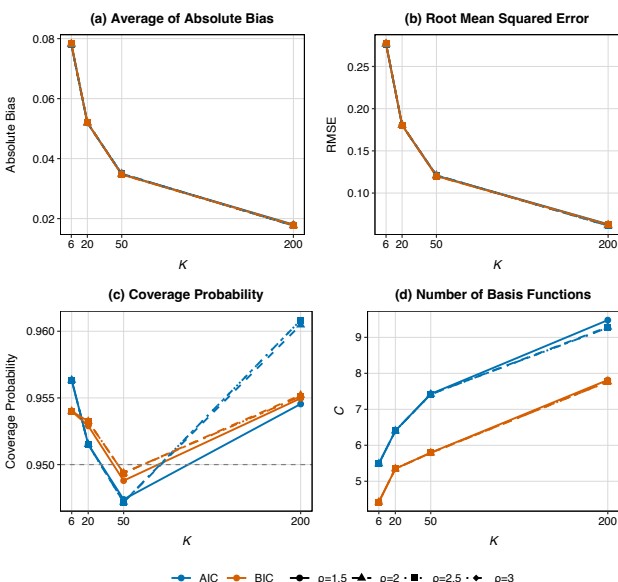

*Figure 4.* Sensitivity analysis of COLSA across different basis selection criteria (AIC vs. BIC) and scaling parameters $\rho \in \{1.5, 2.0, 2.5, 3.0\}$. The blue color indicates the results from AIC and the orange color indicates the result from BIC. Different $\rho$ values are represented by different shapes.

## D. Proof of Main Theorems

This appendix provides complete proofs of Theorem 4.1 (consistency) and Theorem 4.2 (asymptotic normality and oracle efficiency).

We adopt the notation from Section 4. Let $n_j$ denote the sample size of batch $j$ and $N_k := \sum_{j=1}^{k} n_j$ the cumulative sample size. Throughout this appendix, we assume that no single batch dominates the cumulative size, i.e., $n_k = O(N_{k-1})$ for all $k \geq 2$. This ensures that $N_{k-1} \asymp N_k$ as $k \to \infty$. The parameter at degree $p$ is $\theta^{(p)} = (\beta, \gamma^{(p)}) \in \mathbb{R}^{d+p+1}$, where $\beta \in \mathbb{R}^d$ are the regression coefficients and $\gamma^{(p)} \in \mathbb{R}^{p+1}$ are the Bernstein coefficients of the log-baseline hazard. Let $g_0 \in L^2[0, \tau]$ denote the true log-baseline hazard, and let $\Pi_p g_0$ denote its $L^2$-projection onto the space of degree-$p$ Bernstein polynomials. With a slight abuse of notation, we denote $\theta_0^{(p)} = (\beta_0, \Pi_p g_0)$. The following lemma quantifies the approximation error of the sieve.

**Lemma D.1** (Sieve approximation error). *Under condition (C2), the sieve approximation error satisfies:*

$$d(\theta_0^{(p)}, \theta_0) = \underbrace{\|\beta_0 - \beta_0\|}_{=0} + \|\Pi_p g_0 - g_0\|_{L^2} = O(p^{-q}).$$

*Proof.* The result follows from classical Bernstein polynomial approximation theory (Lorentz, 2012): for $g_0 \in \mathcal{S}^q([0, \tau])$, we have $\|g_0 - \Pi_p g_0\|_{L^2} = O(p^{-q})$ by Jackson-type inequalities for polynomial approximation. □

### D.1. Technical Details for Growing Sieve Implementation

This subsection analyzes the error introduced by the shadow mechanism in Section 3.4. Since the sieve degree $p_k = \lfloor CN_k^\nu \rfloor$ grows with the cumulative sample size $N_k$, Hessians from different batches have incompatible dimensions. We show that the working Hessian obtained via the shadow mechanism is asymptotically equivalent to the oracle Hessian.

Let $H_j^{(p)}(\theta)$ denote the Hessian matrix computed from batch $j$ at degree $p$, evaluated at parameter $\theta$. When the argument is omitted, $H_j^{(p)}$ denotes evaluation at $\theta_0^{(p)}$.

The degree elevation matrix $R_{p \to p'} \in \mathbb{R}^{(p'+1) \times (p+1)}$ transforms Bernstein coefficients from degree $p$ to degree $p' > p$ while preserving the polynomial exactly. Its block version is $\bar{R}_{p \to p'} := \text{diag}(I_d, R_{p \to p'}) \in \mathbb{R}^{(d+p'+1) \times (d+p+1)}$, which acts on

the full parameter $\theta^{(p)}$ by leaving $\beta$ unchanged.

The Moore-Penrose inverse is $R^{\dagger} := (R^{\top}R)^{-1}R^{\top}$, satisfying $R^{\dagger}R = I$ since $R$ has full column rank (degree elevation is injective). Recall that the shadow degree is $\bar{p}_j := \lfloor \rho \cdot p_j \rfloor$ with $\rho > 1$. By construction, $R$ satisfies transitivity: $R_{p \to p''} = R_{p' \to p''}R_{p \to p'}$ for $p < p' < p''$. Since the entries of $R_{p \to p'}$ are ratios of binomial coefficients bounded by 1, we have $\|R_{p \to p'}\|_{\mathrm{op}} \le C_R$ for some universal constant $C_R > 0$.

For a Hessian $H^{(p)}$ computed at degree $p$, we define two operations:

$$\mathrm{Embed}_{p \to p'}(H^{(p)}) := \bar{R}_{p \to p'} H^{(p)} (\bar{R}_{p \to p'})^{\top} \in \mathbb{R}^{(d+p'+1) \times (d+p'+1)}, \tag{6}$$

$$\mathrm{Extract}_{p' \to p}(H^{(p')}) := (\bar{R}_{p \to p'})^{\dagger} H^{(p')} ((\bar{R}_{p \to p'})^{\dagger})^{\top} \in \mathbb{R}^{(d+p+1) \times (d+p+1)}. \tag{7}$$

Embedding puts a low-degree Hessian into a higher-dimensional space; extraction retrieves it. These operations satisfy two key properties.

**Lemma D.2** (Properties of embedding). *The embedding operation satisfies:*

(i) *For $p < p'$, we have:*
$$\mathrm{Extract}_{p' \to p}\big(\mathrm{Embed}_{p \to p'}(H^{(p)})\big) = H^{(p)}.$$

(ii) *For $p < p' < p''$, we have:*
$$\mathrm{Embed}_{p' \to p''}\big(\mathrm{Embed}_{p \to p'}(H^{(p)})\big) = \mathrm{Embed}_{p \to p''}(H^{(p)}).$$

*Proof.* (i) By definition (6) and (7):

$$\begin{aligned}
\mathrm{Extract}_{p' \to p}\big(\mathrm{Embed}_{p \to p'}(H^{(p)})\big) &= (\bar{R}_{p \to p'})^{\dagger}\big(\bar{R}_{p \to p'} H^{(p)} \bar{R}_{p \to p'}^{\top}\big)((\bar{R}_{p \to p'})^{\dagger})^{\top} \\
&= ((\bar{R}_{p \to p'})^{\dagger}\bar{R}_{p \to p'})H^{(p)}((\bar{R}_{p \to p'})^{\dagger}\bar{R}_{p \to p'})^{\top} \\
&= I \cdot H^{(p)} \cdot I = H^{(p)},
\end{aligned}$$

where we used $(\bar{R}_{p \to p'})^{\dagger}\bar{R}_{p \to p'} = I$ (left inverse, since $\bar{R}_{p \to p'}$ has full column rank).

(ii) By definition (6) and the transitivity property $R_{p \to p''} = R_{p' \to p''}R_{p \to p'}$:

$$\begin{aligned}
\mathrm{Embed}_{p' \to p''}\big(\mathrm{Embed}_{p \to p'}(H^{(p)})\big) &= \bar{R}_{p' \to p''}\big(\bar{R}_{p \to p'} H^{(p)} \bar{R}_{p \to p'}^{\top}\big)\bar{R}_{p' \to p''}^{\top} \\
&= (\bar{R}_{p' \to p''}\bar{R}_{p \to p'})H^{(p)}(\bar{R}_{p' \to p''}\bar{R}_{p \to p'})^{\top} \\
&= \bar{R}_{p \to p''} H^{(p)} \bar{R}_{p \to p''}^{\top} = \mathrm{Embed}_{p \to p''}(H^{(p)}).
\end{aligned}$$

$\square$

While embedding represents the same matrix in a higher-dimensional space, the chain rule describes how Hessians computed at different degrees on the *same data* are related.

**Lemma D.3** (Hessian chain rule). *For the same batch of data and parameters satisfying $\theta^{(p')} = \bar{R}_{p \to p'}\theta^{(p)}$, the Hessians at degrees $p < p'$ satisfy:*
$$H^{(p)}(\theta^{(p)}) = (\bar{R}_{p \to p'})^{\top} H^{(p')}(\theta^{(p')})\bar{R}_{p \to p'}.$$

In other words, computing the Hessian at higher degree and pulling back via $\bar{R}^{\top} \cdot \bar{R}$ recovers the lower-degree Hessian exactly.

*Proof.* The condition $\theta^{(p')} = \bar{R}_{p \to p'}\theta^{(p)}$ means $(\beta', \gamma') = (\beta, R_{p \to p'}\gamma)$. Since degree elevation preserves the polynomial exactly, both coefficient vectors represent the same Bernstein polynomial:

$$g(t) = \sum_{i=0}^{p} \gamma_i B_{i,p}(t) = \sum_{i=0}^{p'} \gamma_i' B_{i,p'}(t).$$

The log-likelihood $\ell(\theta; \mathcal{O})$ depends on $\theta$ only through the regression coefficients $\beta$ and the function $g(t)$. Since $\beta' = \beta$ and $g(t)$ is the same in both representations, we have $\ell(\theta^{(p)}; \mathcal{O}) = \ell(\theta^{(p')}; \mathcal{O})$. The relation $\theta^{(p')} = \bar{R}_{p \to p'} \theta^{(p)}$ defines a linear reparametrization with Jacobian $J = \bar{R}_{p \to p'}$. By the chain rule for second derivatives of a scalar function $\ell$:

$$\nabla^2_{\theta^{(p)}} \ell = J^\top \nabla^2_{\theta^{(p')}} \ell \cdot J = (\bar{R}_{p \to p'})^\top \nabla^2_{\theta^{(p')}} \ell \cdot \bar{R}_{p \to p'},$$

where the first-derivative correction vanishes since $\bar{R}_{p \to p'}$ is linear. Taking the negative (since $H = -\nabla^2 \ell$) yields the result. $\qquad \square$

We now define the two key quantities for comparing the ideal and practical Hessian aggregation. The **oracle Hessian** is the cumulative Hessian if degree $p_k$ were known from the start:

$$\bar{H}^{*(p_k)}_{k-1} := \sum_{j=1}^{k-1} H_j^{(p_k)}(\theta_0^{(p_k)}).$$

The **working Hessian** is the cumulative Hessian obtained via the shadow mechanism, where each batch $j$ stores $H_j^{(\bar{p}_j)}(\bar{R}_{p_j \to \bar{p}_j} \theta_0^{(p_j)})$ at shadow degree $\bar{p}_j$ and transforms to degree $p_k$ for aggregation. To express the working Hessian explicitly, define the index sets $\mathcal{I}_k := \{j \leq k - 1 : \bar{p}_j < p_k\}$ and $\mathcal{J}_k := \{j \leq k - 1 : \bar{p}_j \geq p_k\}$. Then:

$$\begin{aligned}
\bar{H}^{(p_k)}_{k-1} = &\sum_{j \in \mathcal{I}_k} \bar{R}_{\bar{p}_j \to p_k} H_j^{(\bar{p}_j)}(\bar{R}_{p_j \to \bar{p}_j} \theta_0^{(p_j)}) \bar{R}^\top_{\bar{p}_j \to p_k} \\
&+ \sum_{j \in \mathcal{J}_k} \bar{R}^\top_{p_k \to \bar{p}_j} H_j^{(\bar{p}_j)}(\bar{R}_{p_j \to \bar{p}_j} \theta_0^{(p_j)}) \bar{R}_{p_k \to \bar{p}_j},
\end{aligned}$$

where the first sum uses embedding ($\bar{p}_j < p_k$, shadow insufficient), and the second sum uses the chain rule ($\bar{p}_j \geq p_k$, shadow sufficient).

We now show that the working Hessian is asymptotically equivalent to the oracle Hessian.

**Theorem D.4** (Working Hessian error bound). *Under conditions (C1), (C2), and (C3), with $p_k = \lfloor CN_k^\nu \rfloor$ and $\bar{p}_k = \lfloor \rho \cdot p_k \rfloor$ for $\rho > 1$, there exist constants $C_1, C_2 > 0$ such that:*

$$\|\bar{H}^{*(p_k)}_{k-1} - \bar{H}^{(p_k)}_{k-1}\|_{\mathrm{op}} \leq C_1 \rho^{-1/\nu} N_{k-1} + C_2 N_{k-1} p_k^{1/2-q}. \tag{8}$$

*The first term ($E_{\mathcal{I}}$) arises from batches where $\bar{p}_j < p_k$; the second ($E_{\mathcal{J}}$) from sieve approximation error.*

*Proof.* The difference between the oracle and working Hessians is

$$\bar{H}^{*(p_k)}_{k-1} - \bar{H}^{(p_k)}_{k-1} = \sum_{j=1}^{k-1} H_j^{(p_k)}(\theta_0^{(p_k)}) - \bar{H}^{(p_k)}_{k-1}.$$

We analyze the contributions from $\mathcal{I}_k$ and $\mathcal{J}_k$ separately.

For $j \in \mathcal{J}_k$ (shadow sufficient), since $j < k$ and $N_j < N_k$, we have $p_j \leq p_k \leq \bar{p}_j$. By transitivity, $\bar{R}_{p_j \to \bar{p}_j} = \bar{R}_{p_k \to \bar{p}_j} \bar{R}_{p_j \to p_k}$, so $\bar{R}_{p_j \to \bar{p}_j} \theta_0^{(p_j)} = \bar{R}_{p_k \to \bar{p}_j}(\bar{R}_{p_j \to p_k} \theta_0^{(p_j)})$. The chain rule (Lemma D.3) then gives

$$\bar{R}^\top_{p_k \to \bar{p}_j} H_j^{(\bar{p}_j)}(\bar{R}_{p_j \to \bar{p}_j} \theta_0^{(p_j)}) \bar{R}_{p_k \to \bar{p}_j} = H_j^{(p_k)}(\bar{R}_{p_j \to p_k} \theta_0^{(p_j)}).$$

Thus, the working Hessian contribution from batch $j$ equals $H_j^{(p_k)}(\bar{R}_{p_j \to p_k} \theta_0^{(p_j)})$, which differs from the oracle term $H_j^{(p_k)}(\theta_0^{(p_k)})$ only in the evaluation point.

For $j \in \mathcal{I}_k$ (shadow insufficient), the chain rule does not apply since $\bar{p}_j < p_k$. The working Hessian contribution is the embedded matrix $\bar{R}_{\bar{p}_j \to p_k} H_j^{(\bar{p}_j)}(\bar{R}_{p_j \to \bar{p}_j} \theta_0^{(p_j)}) \bar{R}^\top_{\bar{p}_j \to p_k}$.

Combining these observations, the error decomposes as

$$\bar{H}_{k-1}^{*(p_k)} - \bar{H}_{k-1}^{(p_k)} = \underbrace{\sum_{j \in \mathcal{I}_k} \left( H_j^{(p_k)}(\theta_0^{(p_k)}) - \bar{R}_{\bar{p}_j \to p_k} H_j^{(\bar{p}_j)}(\bar{R}_{p_j \to \bar{p}_j} \theta_0^{(p_j)}) \bar{R}_{\bar{p}_j \to p_k}^\top \right)}_{E_\mathcal{I}}$$

$$+ \underbrace{\sum_{j \in \mathcal{J}_k} \left( H_j^{(p_k)}(\theta_0^{(p_k)}) - H_j^{(p_k)}(\bar{R}_{p_j \to p_k} \theta_0^{(p_j)}) \right)}_{E_\mathcal{J}}.$$

To bound $E_\mathcal{I}$, we use the triangle inequality. Since $\bar{p}_j < p_k$ implies $p_j < p_k/\rho$, each term satisfies

$$\|H_j^{(p_k)}(\theta_0^{(p_k)}) - \bar{R}_{\bar{p}_j \to p_k} H_j^{(\bar{p}_j)}(\bar{R}_{p_j \to \bar{p}_j} \theta_0^{(p_j)}) \bar{R}_{\bar{p}_j \to p_k}^\top\|_{\mathrm{op}} \leq \|H_j^{(p_k)}(\theta_0^{(p_k)})\|_{\mathrm{op}} + C_R^2 \|H_j^{(\bar{p}_j)}(\bar{R}_{p_j \to \bar{p}_j} \theta_0^{(p_j)})\|_{\mathrm{op}}.$$

By conditions (C1) and (C3), the per-observation Hessian is bounded: $\|\nabla_\theta^2 \ell_{j,i}(\theta)\|_{\mathrm{op}} \leq M$ for all $j, i, \theta$. Thus, $\|H_j^{(p)}\|_{\mathrm{op}} \leq M n_j$ for any degree $p$, and:

$$\|E_\mathcal{I}\|_{\mathrm{op}} \leq (1 + C_R^2) M \sum_{j \in \mathcal{I}_k} n_j. \tag{9}$$

We now bound $\sum_{j \in \mathcal{I}_k} n_j / N_{k-1}$. By definition, $j \in \mathcal{I}_k$ means $\bar{p}_j < p_k$. Since $\bar{p}_j = \lfloor \rho p_j \rfloor \geq \rho p_j - 1$, the condition $\bar{p}_j < p_k$ implies

$$\rho p_j - 1 < p_k, \quad \text{i.e.,} \quad p_j < \frac{p_k + 1}{\rho}.$$

Recall that $p_j = \lfloor CN_j^\nu \rfloor \geq CN_j^\nu - 1$ and $p_k = \lfloor CN_k^\nu \rfloor \leq CN_k^\nu$. Combining these inequalities:

$$CN_j^\nu - 1 < \frac{CN_k^\nu + 1}{\rho}, \quad \text{hence} \quad N_j < \left( \frac{N_k^\nu + 1/C}{\rho} + \frac{1}{C} \right)^{1/\nu}.$$

For large $k$, this simplifies to $N_j \leq c_\rho N_k$ for some constant $c_\rho := \rho^{-1/\nu} < 1$. Therefore, $\mathcal{I}_k \subseteq \{j : N_j \leq c_\rho N_k\}$, and:

$$\frac{\sum_{j \in \mathcal{I}_k} n_j}{N_{k-1}} \leq \frac{c_\rho N_k}{N_{k-1}} \to c_\rho \quad \text{as } k \to \infty. \tag{10}$$

To bound $E_\mathcal{J}$, note that the error arises from evaluating the Hessian at $\bar{R}_{p_j \to p_k} \theta_0^{(p_j)}$ instead of $\theta_0^{(p_k)}$. When $p_j = p_k$, we have $\bar{R}_{p_j \to p_k} = I$, so the corresponding term vanishes. When $p_j < p_k$, the parameter $\bar{R}_{p_j \to p_k} \theta_0^{(p_j)} = (\beta_0, R_{p_j \to p_k} \gamma_0^{(p_j)})$ represents $\Pi_{p_j} g_0$ in the degree-$p_k$ basis, which differs from $\theta_0^{(p_k)} = (\beta_0, \gamma_0^{(p_k)})$ representing $\Pi_{p_k} g_0$. By the Lipschitz continuity of the per-observation Hessian (conditions (C1) and (C3)), there exists $L > 0$ such that:

$$\|H_j^{(p_k)}(\theta_0^{(p_k)}) - H_j^{(p_k)}(\bar{R}_{p_j \to p_k} \theta_0^{(p_j)})\|_{\mathrm{op}} \leq L n_j \|\theta_0^{(p_k)} - \bar{R}_{p_j \to p_k} \theta_0^{(p_j)}\|.$$

By the norm equivalence for Bernstein polynomials (Lorentz, 2012), we have $\|\gamma\| \asymp \sqrt{p} \|g\|_{L^2}$, so:

$$\|\theta_0^{(p_k)} - \bar{R}_{p_j \to p_k} \theta_0^{(p_j)}\| = \|\gamma_0^{(p_k)} - R_{p_j \to p_k} \gamma_0^{(p_j)}\| \lesssim \sqrt{p_k} \|\Pi_{p_k} g_0 - \Pi_{p_j} g_0\|_{L^2}.$$

By condition (C2), $\|g_0 - \Pi_p g_0\|_{L^2} = O(p^{-q})$. Since $\Pi_{p_k} g_0 - \Pi_{p_j} g_0$ is orthogonal to $g_0 - \Pi_{p_k} g_0$, we have $\|\Pi_{p_k} g_0 - \Pi_{p_j} g_0\|_{L^2} \leq \|g_0 - \Pi_{p_j} g_0\|_{L^2} = O(p_j^{-q})$.

For $j \in \mathcal{J}_k$ with $p_j < p_k$, we have $\bar{p}_j \geq p_k$, so $p_j \geq p_k/\rho$, giving $p_j^{-q} \leq (\rho/p_k)^q = O(p_k^{-q})$. Therefore:

$$\|E_\mathcal{J}\|_{\mathrm{op}} \leq \sum_{j \in \mathcal{J}_k, p_j < p_k} L n_j \cdot O(\sqrt{p_k} \cdot p_k^{-q}) = O(N_{k-1} \cdot p_k^{1/2 - q}). \tag{11}$$

Combining (9) and (10), we obtain $\|E_\mathcal{I}\|_{\mathrm{op}} \leq (1 + C_R^2) M \rho^{-1/\nu} N_{k-1}$. Together with (11):

$$\|\bar{H}_{k-1}^{*(p_k)} - \bar{H}_{k-1}^{(p_k)}\|_{\mathrm{op}} \leq (1 + C_R^2) M \rho^{-1/\nu} N_{k-1} + O(N_{k-1} \cdot p_k^{1/2 - q}).$$

For $q \geq 2$ as assumed in (C2), we have $1/2 - q \leq -3/2 < 0$, so $E_\mathcal{J} = O(N_{k-1} \cdot p_k^{1/2-q}) = o(N_{k-1})$. $\qquad \square$

## D.2. Proof of Theorem 4.1

We establish consistency by showing that the renewable aggregated estimator $\tilde{\theta}_k^{(p_k)}$ converges to $\theta_0$ in the metric $d(\cdot, \cdot)$. Following the sieve estimation framework of Shen & Wong (1994), adapted to the renewable setting of Luo & Song (2020), we verify the required regularity conditions.

*Proof.* We prove by induction on $k$ that for all $k \geq 1$:

$$d(\tilde{\theta}_k^{(p_k)}, \theta_0) = O_p(N_k^{-\min(q\nu,(1-\nu)/2)}). \tag{12}$$

**Base case ($k = 1$):** For the first batch, $\tilde{\theta}_1^{(p_1)}$ is the standard sieve MLE. By Tang et al. (2023):

$$d(\tilde{\theta}_1^{(p_1)}, \theta_0) = O_p(n_1^{-\min(q\nu,(1-\nu)/2)}).$$

**Inductive step:** Assume (12) holds for $k - 1$:

$$d(\tilde{\theta}_{k-1}^{(p_{k-1})}, \theta_0) = O_p(N_{k-1}^{-\min(q\nu,(1-\nu)/2)}).$$

We show that (12) holds for $k$. Let $L_k(\theta) := \sum_{j=1}^k \ell_j(\theta)$ denote the cumulative log-likelihood. The oracle sieve MLE is $\theta_k^* := \arg\max_{\theta \in \Theta_N^{p_k}} L_k(\theta)$. Our renewable estimator $\tilde{\theta}_k^{(p_k)}$ instead maximizes the surrogate likelihood:

$$\tilde{L}_k(\theta) = \ell_k(\theta) + \sum_{j=1}^{k-1} \ell_j(\tilde{\theta}_{k-1}^{(p_k)}) - \frac{1}{2}(\theta - \tilde{\theta}_{k-1}^{(p_k)})^\top \tilde{H}_{k-1}^{(p_k)}(\theta - \tilde{\theta}_{k-1}^{(p_k)}),$$

Here $\tilde{\theta}_{k-1}^{(p_k)} := \bar{R}_{p_{k-1} \to p_k} \tilde{\theta}_{k-1}^{(p_{k-1})}$ denotes the degree-elevated estimate obtained by applying the degree elevation matrix to the previous estimate, and $\tilde{H}_{k-1}^{(p_k)}$ is the working Hessian obtained via the shadow mechanism (see Appendix D.1):

$$\tilde{H}_{k-1}^{(p_k)} := \sum_{j \in \mathcal{I}_k} \bar{R}_{\bar{p}_j \to p_k} H_j^{(\bar{p}_j)}(\tilde{\theta}_j^{(\bar{p}_j)}) \bar{R}_{\bar{p}_j \to p_k}^\top + \sum_{j \in \mathcal{J}_k} \bar{R}_{p_k \to \bar{p}_j}^\top H_j^{(\bar{p}_j)}(\tilde{\theta}_j^{(\bar{p}_j)}) \bar{R}_{p_k \to \bar{p}_j},$$

where $\mathcal{I}_k := \{j \leq k-1 : \bar{p}_j < p_k\}$ (shadow insufficient, use embedding), $\mathcal{J}_k := \{j \leq k-1 : \bar{p}_j \geq p_k\}$ (shadow sufficient, use chain rule), and $\bar{p}_j = \lfloor \rho \cdot p_j \rfloor$ is the shadow degree. Note that $\tilde{\theta}_j^{(\bar{p}_j)} := \bar{R}_{p_j \to \bar{p}_j} \tilde{\theta}_j^{(p_j)}$ is the degree-elevated estimate at the shadow degree.

The analysis in Tang et al. (2023) establishes the convergence rate for the *exact* sieve MLE $\theta_k^*$, which maximizes $L_k(\theta)$ over $\Theta_N^{p_k}$. Our renewable estimator $\tilde{\theta}_k^{(p_k)}$ instead maximizes the surrogate likelihood $\tilde{L}_k$, and hence is only an *approximate* maximizer of $L_k$. To handle this, we invoke the approximate maximizer framework of Shen & Wong (1994): if an estimator $\hat{\theta}$ satisfies

$$L_k(\hat{\theta}) \geq \sup_{\theta \in \Theta_N^{p_k}} L_k(\theta) - \eta_n, \quad \text{with } \eta_n = o(N_k^{1-2\nu}), \tag{13}$$

then the same convergence rate as the exact maximizer is preserved. Once we verify that $\tilde{\theta}_k^{(p_k)}$ satisfies (13), the remainder of the convergence rate analysis follows exactly as in Tang et al. (2023) (Theorem 1), since it depends only on the sieve approximation properties and regularity conditions, not on exact maximization.

We now verify (13) by bounding $L_k(\theta_k^*) - L_k(\tilde{\theta}_k^{(p_k)})$.

Decompose the gap as:

$$L_k(\theta_k^*) - L_k(\tilde{\theta}_k^{(p_k)}) = \underbrace{[L_k(\theta_k^*) - \tilde{L}_k(\theta_k^*)]}_{(I)} + \underbrace{[\tilde{L}_k(\theta_k^*) - \tilde{L}_k(\tilde{\theta}_k^{(p_k)})]}_{\leq 0}$$
$$+ \underbrace{[\tilde{L}_k(\tilde{\theta}_k^{(p_k)}) - L_k(\tilde{\theta}_k^{(p_k)})]}_{(III)}.$$

The middle term is non-positive since $\tilde{\theta}_k^{(p_k)}$ maximizes $\tilde{L}_k$ over $\Theta_N^{p_k}$.

To bound term (I), we first expand $L_k(\theta_k^*) - \tilde{L}_k(\theta_k^*)$ using the definitions. Since $L_k = \ell_k + \sum_{j=1}^{k-1} \ell_j$ and $\tilde{L}_k(\theta) = \ell_k(\theta) + \sum_{j=1}^{k-1} \ell_j(\tilde{\theta}_{k-1}^{(p_k)}) - \frac{1}{2}(\theta - \tilde{\theta}_{k-1}^{(p_k)})^\top \tilde{H}_{k-1}^{(p_k)}(\theta - \tilde{\theta}_{k-1}^{(p_k)})$, we have:

$$
\begin{aligned}
(I) &= L_k(\theta_k^*) - \tilde{L}_k(\theta_k^*) \\
&= \sum_{j=1}^{k-1} \ell_j(\theta_k^*) - \sum_{j=1}^{k-1} \ell_j(\tilde{\theta}_{k-1}^{(p_k)}) + \frac{1}{2}(\theta_k^* - \tilde{\theta}_{k-1}^{(p_k)})^\top \tilde{H}_{k-1}^{(p_k)}(\theta_k^* - \tilde{\theta}_{k-1}^{(p_k)}).
\end{aligned}
$$

Now apply second-order Taylor expansion to the historical likelihood $\sum_{j=1}^{k-1} \ell_j(\theta_k^*)$ around $\tilde{\theta}_{k-1}^{(p_k)}$:

$$
\begin{aligned}
\sum_{j=1}^{k-1} \ell_j(\theta_k^*) - \sum_{j=1}^{k-1} \ell_j(\tilde{\theta}_{k-1}^{(p_k)}) &= \left( \sum_{j=1}^{k-1} \nabla \ell_j(\tilde{\theta}_{k-1}^{(p_k)}) \right)^\top (\theta_k^* - \tilde{\theta}_{k-1}^{(p_k)}) \\
&\quad - \frac{1}{2}(\theta_k^* - \tilde{\theta}_{k-1}^{(p_k)})^\top \left( \sum_{j=1}^{k-1} H_j^{(p_k)}(\xi) \right) (\theta_k^* - \tilde{\theta}_{k-1}^{(p_k)}),
\end{aligned}
$$

where $\xi$ lies on the line segment between $\tilde{\theta}_{k-1}^{(p_k)}$ and $\theta_k^*$.

Substituting into (I):

$$
(I) = \left( \sum_{j=1}^{k-1} \nabla \ell_j(\tilde{\theta}_{k-1}^{(p_k)}) \right)^\top (\theta_k^* - \tilde{\theta}_{k-1}^{(p_k)}) + \frac{1}{2}(\theta_k^* - \tilde{\theta}_{k-1}^{(p_k)})^\top \left( \tilde{H}_{k-1}^{(p_k)} - \sum_{j=1}^{k-1} H_j^{(p_k)}(\xi) \right) (\theta_k^* - \tilde{\theta}_{k-1}^{(p_k)}).
$$

We now bound each term using the inductive hypothesis. To bound $\|\theta_k^* - \tilde{\theta}_{k-1}^{(p_k)}\|$, we first bound $d(\theta_k^*, \tilde{\theta}_{k-1}^{(p_k)})$ via the triangle inequality:

$$
d(\theta_k^*, \tilde{\theta}_{k-1}^{(p_k)}) \leq d(\theta_k^*, \theta_0^{(p_k)}) + d(\theta_0^{(p_k)}, \theta_0) + d(\theta_0, \tilde{\theta}_{k-1}^{(p_k)}).
$$

We bound each term separately. The first term satisfies $d(\theta_k^*, \theta_0^{(p_k)}) = O_p(N_k^{-\min(q\nu, (1-\nu)/2)})$ by Tang et al. (2023). The second term is bounded by Lemma D.1:

$$
d(\theta_0^{(p_k)}, \theta_0) = O(p_k^{-q}) = O(N_k^{-q\nu}).
$$

For the third term, since degree elevation preserves $(\beta, g)$, we have $d(\theta_0, \tilde{\theta}_{k-1}^{(p_k)}) = d(\theta_0, \tilde{\theta}_{k-1}^{(p_{k-1})})$. By the inductive hypothesis:

$$
d(\theta_0, \tilde{\theta}_{k-1}^{(p_k)}) = O_p(N_{k-1}^{-\min(q\nu, (1-\nu)/2)}).
$$

Since $N_k \geq N_{k-1}$ and $q\nu \geq \min(q\nu, (1-\nu)/2)$, all three terms are $O_p(N_{k-1}^{-\min(q\nu, (1-\nu)/2)})$, and hence $d(\theta_k^*, \tilde{\theta}_{k-1}^{(p_k)}) = O_p(N_{k-1}^{-\min(q\nu, (1-\nu)/2)})$. By the norm equivalence $\|\gamma\| \asymp \sqrt{p}\|g\|_{L^2}$ (Lorentz, 2012):

$$
\|\theta_k^* - \tilde{\theta}_{k-1}^{(p_k)}\| = O_p\left(\sqrt{p_k} \cdot N_{k-1}^{-\min(q\nu, (1-\nu)/2)}\right). \tag{14}
$$

For the gradient term, we decompose the sum at the random point $\tilde{\theta}_{k-1}^{(p_k)}$ using the true parameter $\theta_0^{(p_k)}$:

$$
\sum_{j=1}^{k-1} \nabla \ell_j(\tilde{\theta}_{k-1}^{(p_k)}) = \sum_{j=1}^{k-1} \nabla \ell_j(\theta_0^{(p_k)}) + \sum_{j=1}^{k-1} \left[ \nabla \ell_j(\tilde{\theta}_{k-1}^{(p_k)}) - \nabla \ell_j(\theta_0^{(p_k)}) \right].
$$

For the first sum, since $\{\nabla \ell_j(\theta_0^{(p_k)})\}_{j=1}^{k-1}$ are independent across batches with $E[\nabla \ell_j(\theta_0^{(p_k)})] = O(p_k^{-q})$ (the bias from evaluating at the projected parameter), the CLT for independent random vectors gives $\| \sum_{j=1}^{k-1} \nabla \ell_j(\theta_0^{(p_k)}) \| = O_p(\sqrt{N_{k-1} p_k})$;

the Lindeberg condition holds uniformly over $p_k$ by conditions (C1)–(C3). For the second sum, by the mean value theorem and the Lipschitz continuity of the score function (conditions (C1) and (C3)):

$$\left\| \sum_{j=1}^{k-1} \left[ \nabla \ell_j(\tilde{\theta}_{k-1}^{(p_k)}) - \nabla \ell_j(\theta_0^{(p_k)}) \right] \right\| \lesssim N_{k-1} \| \tilde{\theta}_{k-1}^{(p_k)} - \theta_0^{(p_k)} \|.$$

By the same argument as (14), $\| \tilde{\theta}_{k-1}^{(p_k)} - \theta_0^{(p_k)} \|$ is of the same order. Hence:

$$\left\| \sum_{j=1}^{k-1} \nabla \ell_j(\tilde{\theta}_{k-1}^{(p_k)}) \right\| = O_p(\sqrt{N_{k-1} p_k}) + O_p(N_{k-1}^{1-\min(q\nu,(1-\nu)/2)} \sqrt{p_k}).$$

By Cauchy-Schwarz and (14):

$$\left| \left( \sum_{j=1}^{k-1} \nabla \ell_j(\tilde{\theta}_{k-1}^{(p_k)}) \right)^\top (\theta_k^* - \tilde{\theta}_{k-1}^{(p_k)}) \right| = O_p \left( p_k \cdot N_{k-1}^{1/2-\min(q\nu,(1-\nu)/2)} \right) + O_p \left( p_k \cdot N_{k-1}^{1-2\min(q\nu,(1-\nu)/2)} \right).$$

Since $p_k = O(N_k^\nu)$ and $N_{k-1} \asymp N_k$, the two terms are $O_p(N_k^{\nu+1/2-\min(q\nu,(1-\nu)/2)})$ and $O_p(N_k^{\nu+1-2\min(q\nu,(1-\nu)/2)})$, respectively. Both are $o(N_k^{1-2\nu})$ provided $\min(q\nu,(1-\nu)/2) > 3\nu/2$, which holds under $\nu < 1/(2q)$ for $q \geq 2$: when $\min = q\nu$, we need $q > 3/2$; when $\min = (1-\nu)/2$, we need $\nu < 1/4$.

For the Hessian mismatch term, by the triangle inequality:

$$\left\| \tilde{H}_{k-1}^{(p_k)} - \sum_{j=1}^{k-1} H_j^{(p_k)}(\xi) \right\|_{\text{op}} \leq \underbrace{\| \tilde{H}_{k-1}^{(p_k)} - \bar{H}_{k-1}^{(p_k)} \|_{\text{op}}}_{(A_1)} + \underbrace{\| \bar{H}_{k-1}^{(p_k)} - \bar{H}_{k-1}^{*(p_k)} \|_{\text{op}}}_{(A_2)}$$
$$+ \underbrace{\left\| \bar{H}_{k-1}^{*(p_k)} - \sum_{j=1}^{k-1} H_j^{(p_k)}(\xi) \right\|_{\text{op}}}_{(A_3)}.$$

For $(A_1)$, note that $\tilde{H}_{k-1}^{(p_k)}$ aggregates $H_j^{(\bar{p}_j)}(\tilde{\theta}_j^{(\bar{p}_j)})$, while $\bar{H}_{k-1}^{(p_k)}$ uses $H_j^{(\bar{p}_j)}(\bar{R}_{p_j \to \bar{p}_j} \theta_0^{(p_j)})$. By conditions (C1) and (C3), the Hessian is Lipschitz: $\| H_j^{(p)}(\theta) - H_j^{(p)}(\theta') \|_{\text{op}} \leq C n_j \| \theta - \theta' \|$ for some constant $C > 0$. By norm equivalence, $\| \tilde{\theta}_j^{(\bar{p}_j)} - \bar{R}_{p_j \to \bar{p}_j} \theta_0^{(p_j)} \| \lesssim \sqrt{\bar{p}_j} \cdot d(\tilde{\theta}_j^{(\bar{p}_j)}, \theta_0)$. Combining with the inductive hypothesis $d(\tilde{\theta}_j^{(\bar{p}_j)}, \theta_0) = O_p(N_j^{-\min(q\nu,(1-\nu)/2)})$:

$$\| H_j^{(\bar{p}_j)}(\tilde{\theta}_j^{(\bar{p}_j)}) - H_j^{(\bar{p}_j)}(\bar{R}_{p_j \to \bar{p}_j} \theta_0^{(p_j)}) \|_{\text{op}} = O_p \left( n_j \sqrt{\bar{p}_j} \cdot N_j^{-\min(q\nu,(1-\nu)/2)} \right).$$

Since $\bar{p}_j \leq p_k$ and the embedding/extraction operators have bounded operator norm (bounded by $C_R^2$ as established in Section D.1), summing over $j$ gives:

$$(A_1) \leq \sqrt{p_k} \sum_{j=1}^{k-1} n_j N_j^{-\min(q\nu,(1-\nu)/2)} = O_p \left( N_{k-1}^{1+\nu/2-\min(q\nu,(1-\nu)/2)} \right),$$

where we used $\sum_{j=1}^{k-1} n_j N_j^{-r} = O(N_{k-1}^{1-r})$ for $r \in (0,1)$.

For $(A_2)$, by (8):

$$(A_2) \leq C_1 \rho^{-1/\nu} N_{k-1} + C_2 N_{k-1} p_k^{1/2-q}.$$

For $(A_3)$, since $\xi$ lies on the segment between $\tilde{\theta}_{k-1}^{(p_k)}$ and $\theta_k^*$, we have $\| \xi - \theta_0^{(p_k)} \| = O_p(\sqrt{p_k} N_{k-1}^{-\min(q\nu,(1-\nu)/2)})$, and by Lipschitz continuity:

$$(A_3) = O_p(N_{k-1}^{1+\nu/2-\min(q\nu,(1-\nu)/2)}).$$

We now bound the quadratic form. By (14), $\|\theta_k^* - \tilde{\theta}_{k-1}^{(p_k)}\|^2 = O_p(p_k N_{k-1}^{-2\min(q\nu,(1-\nu)/2)})$. Multiplying by the bounds on $(A_1)$, $(A_2)$, and $(A_3)$, each contribution is $o(N_k^{1-2\nu})$ under $\nu < 1/(2q)$ for $q \geq 2$, since this condition implies $\min(q\nu, (1-\nu)/2) > 3\nu/2$. In particular, the $E_{\mathcal{I}}$ contribution is $O_p(\rho^{-1/\nu} N_k^{1-3\nu}) = o(N_k^{1-2\nu})$ for any fixed $\rho > 1$.

Combining these bounds, we obtain $(I) = o(N_k^{1-2\nu})$.

To bound term (III), recall that $(III) = \tilde{L}_k(\tilde{\theta}_k^{(p_k)}) - L_k(\tilde{\theta}_k^{(p_k)})$. By the same expansion as for term (I):

$$(III) = -\sum_{j=1}^{k-1} \ell_j(\tilde{\theta}_k^{(p_k)}) + \sum_{j=1}^{k-1} \ell_j(\tilde{\theta}_{k-1}^{(p_k)}) - \frac{1}{2}(\tilde{\theta}_k^{(p_k)} - \tilde{\theta}_{k-1}^{(p_k)})^\top \tilde{H}_{k-1}^{(p_k)}(\tilde{\theta}_k^{(p_k)} - \tilde{\theta}_{k-1}^{(p_k)}).$$

We first establish a bound on $\|\tilde{\theta}_k^{(p_k)} - \tilde{\theta}_{k-1}^{(p_k)}\|$. Recall that $\tilde{L}_k(\theta) = \ell_k(\theta) - \frac{1}{2}(\theta - \tilde{\theta}_{k-1}^{(p_k)})^\top \tilde{H}_{k-1}^{(p_k)}(\theta - \tilde{\theta}_{k-1}^{(p_k)}) + \text{const}$, so

$$\nabla \tilde{L}_k(\theta) = \nabla \ell_k(\theta) - \tilde{H}_{k-1}^{(p_k)}(\theta - \tilde{\theta}_{k-1}^{(p_k)}).$$

The first-order condition $\nabla \tilde{L}_k(\tilde{\theta}_k^{(p_k)}) = 0$ yields:

$$\nabla \ell_k(\tilde{\theta}_k^{(p_k)}) = \tilde{H}_{k-1}^{(p_k)}(\tilde{\theta}_k^{(p_k)} - \tilde{\theta}_{k-1}^{(p_k)}).$$

Expanding $\nabla \ell_k(\tilde{\theta}_k^{(p_k)})$ around $\tilde{\theta}_{k-1}^{(p_k)}$ using $\nabla \ell_k(\tilde{\theta}_k) = \nabla \ell_k(\tilde{\theta}_{k-1}) - H_k(\xi)(\tilde{\theta}_k - \tilde{\theta}_{k-1})$ (since $H_k = -\nabla^2 \ell_k$) and substituting:

$$\nabla \ell_k(\tilde{\theta}_{k-1}^{(p_k)}) = (\tilde{H}_{k-1}^{(p_k)} + H_k^{(p_k)}(\xi))(\tilde{\theta}_k^{(p_k)} - \tilde{\theta}_{k-1}^{(p_k)})$$

for some $\xi$ on the segment between $\tilde{\theta}_{k-1}^{(p_k)}$ and $\tilde{\theta}_k^{(p_k)}$. To bound $\|\tilde{\theta}_k^{(p_k)} - \tilde{\theta}_{k-1}^{(p_k)}\|$, we need the matrix $\tilde{H}_{k-1}^{(p_k)} + H_k^{(p_k)}(\xi)$ to be invertible with controlled inverse norm. Since both $\tilde{H}_{k-1}^{(p_k)}$ and $H_k^{(p_k)}(\xi)$ are positive semidefinite with $\|\tilde{H}_{k-1}^{(p_k)}\| = O(N_{k-1})$, this reduces to establishing a lower bound on $\lambda_{\min}(\tilde{H}_{k-1}^{(p_k)})$.

By condition (C6), $\lambda_{\min}(\bar{H}_{k-1}^{*(p_k)}) \geq \underline{\lambda} N_{k-1}$. From the bounds on $(A_1)$ and $(A_2)$ in term (I), $\|\tilde{H}_{k-1}^{(p_k)} - \bar{H}_{k-1}^{*(p_k)}\|_{\text{op}} \leq (A_1) + E_{\mathcal{I}} + E_{\mathcal{J}}$, where $(A_1) + E_{\mathcal{J}} = o(N_{k-1})$ and $E_{\mathcal{I}} = C_1 \rho^{-1/\nu} N_{k-1}$. For $\rho \geq (4C_1/\underline{\lambda})^\nu$ and $N_{k-1}$ sufficiently large, by Weyl's inequality:

$$\lambda_{\min}(\tilde{H}_{k-1}^{(p_k)}) \geq \frac{\underline{\lambda}}{2} N_{k-1}, \quad \text{hence} \quad \lambda_{\min}(\tilde{H}_{k-1}^{(p_k)} + H_k^{(p_k)}(\xi)) \geq \frac{\underline{\lambda}}{2} N_{k-1}.$$

It remains to bound $\|\nabla \ell_k(\tilde{\theta}_{k-1}^{(p_k)})\|$. By the triangle inequality:

$$\|\nabla \ell_k(\tilde{\theta}_{k-1}^{(p_k)})\| \leq \|\nabla \ell_k(\theta_0^{(p_k)})\| + \|\nabla \ell_k(\tilde{\theta}_{k-1}^{(p_k)}) - \nabla \ell_k(\theta_0^{(p_k)})\|.$$

The first term is $O_p(\sqrt{n_k p_k})$ by the CLT for the score function. For the second term, by the Lipschitz continuity of the score (conditions (C1) and (C3)) and the inductive hypothesis $\|\tilde{\theta}_{k-1}^{(p_k)} - \theta_0^{(p_k)}\| = O_p(\sqrt{p_k} N_{k-1}^{-\min(q\nu,(1-\nu)/2)})$:

$$\|\nabla \ell_k(\tilde{\theta}_{k-1}^{(p_k)}) - \nabla \ell_k(\theta_0^{(p_k)})\| = O_p(n_k \sqrt{p_k} N_{k-1}^{-\min(q\nu,(1-\nu)/2)}) = o_p(\sqrt{n_k p_k})$$

under $\nu < 1/(2q)$, since

$$n_k \sqrt{p_k} N_{k-1}^{-\min(q\nu,(1-\nu)/2)} / \sqrt{n_k p_k} = \sqrt{n_k} N_{k-1}^{-\min(q\nu,(1-\nu)/2)} \to 0.$$

Thus, $\|\nabla \ell_k(\tilde{\theta}_{k-1}^{(p_k)})\| = O_p(\sqrt{n_k p_k})$, and:

$$\|\tilde{\theta}_k^{(p_k)} - \tilde{\theta}_{k-1}^{(p_k)}\| = O_p(N_{k-1}^{-1} \sqrt{n_k p_k}) = O_p(N_{k-1}^{-1} \sqrt{n_k} N_k^{\nu/2}).$$

With $n_k \leq N_k$ and $N_{k-1} \asymp N_k$, this gives:

$$\|\tilde{\theta}_k^{(p_k)} - \tilde{\theta}_{k-1}^{(p_k)}\| = O_p(N_k^{-(1-\nu)/2}). \tag{15}$$

Now we bound $(III)$ using Taylor expansion. By a second-order expansion analogous to that used for term (I):

$$\sum_{j=1}^{k-1} \ell_j(\tilde{\theta}_k^{(p_k)}) - \sum_{j=1}^{k-1} \ell_j(\tilde{\theta}_{k-1}^{(p_k)}) = \left(\sum_{j=1}^{k-1} \nabla \ell_j(\tilde{\theta}_{k-1}^{(p_k)})\right)^\top (\tilde{\theta}_k^{(p_k)} - \tilde{\theta}_{k-1}^{(p_k)})$$

$$- \frac{1}{2}(\tilde{\theta}_k^{(p_k)} - \tilde{\theta}_{k-1}^{(p_k)})^\top \left(\sum_{j=1}^{k-1} H_j^{(p_k)}(\zeta)\right) (\tilde{\theta}_k^{(p_k)} - \tilde{\theta}_{k-1}^{(p_k)}),$$

where $\zeta$ lies on the segment between $\tilde{\theta}_{k-1}^{(p_k)}$ and $\tilde{\theta}_k^{(p_k)}$. Substituting into the expression for $(III)$:

$$(III) = -\left(\sum_{j=1}^{k-1} \nabla \ell_j(\tilde{\theta}_{k-1}^{(p_k)})\right)^\top (\tilde{\theta}_k^{(p_k)} - \tilde{\theta}_{k-1}^{(p_k)})$$

$$+ \frac{1}{2}(\tilde{\theta}_k^{(p_k)} - \tilde{\theta}_{k-1}^{(p_k)})^\top \left(\sum_{j=1}^{k-1} H_j^{(p_k)}(\zeta) - \tilde{H}_{k-1}^{(p_k)}\right) (\tilde{\theta}_k^{(p_k)} - \tilde{\theta}_{k-1}^{(p_k)}).$$

For the gradient term, using the decomposition established earlier and (15):

$$\left|\left(\sum_{j=1}^{k-1} \nabla \ell_j(\tilde{\theta}_{k-1}^{(p_k)})\right)^\top (\tilde{\theta}_k^{(p_k)} - \tilde{\theta}_{k-1}^{(p_k)})\right| = O_p\left(\sqrt{N_{k-1}p_k} \cdot N_k^{-1/2+\nu/2}\right) = O_p(N_k^\nu).$$

Since $\nu < 1/(2q) \leq 1/4$ for $q \geq 2$, we have $\nu < 1 - 2\nu$, so this is $o(N_k^{1-2\nu})$.

For the Hessian mismatch term, we bound the quadratic form:

$$\left|(\tilde{\theta}_k - \tilde{\theta}_{k-1})^\top \left(\sum_{j=1}^{k-1} H_j(\zeta) - \tilde{H}_{k-1}\right) (\tilde{\theta}_k - \tilde{\theta}_{k-1})\right| \leq \|\tilde{\theta}_k - \tilde{\theta}_{k-1}\|^2 \cdot \left\|\sum_{j=1}^{k-1} H_j(\zeta) - \tilde{H}_{k-1}\right\|_{op}.$$

From (15), $\|\tilde{\theta}_k - \tilde{\theta}_{k-1}\|^2 = O_p(N_k^{-(1-\nu)})$. For the operator norm, by the same triangle inequality decomposition as in term (I):

$$\left\|\sum_{j=1}^{k-1} H_j(\zeta) - \tilde{H}_{k-1}\right\|_{op} \leq (A_1) + (A_2) + (A_3'),$$

where $(A_3')$ is analogous to $(A_3)$ with $\zeta$ in place of $\xi$. Since $\zeta$ lies between $\tilde{\theta}_{k-1}$ and $\tilde{\theta}_k$, $(A_3')$ has the same order as $(A_1)$. The dominant contributions come from $(A_1)$ and $E_{\mathcal{I}}$: multiplying by $(A_1) = O_p(N_k^{1+\nu/2-\min(q\nu,(1-\nu)/2)})$ gives $O_p(N_k^{3\nu/2-\min(q\nu,(1-\nu)/2)})$, while multiplying by $E_{\mathcal{I}} = O(\rho^{-1/\nu}N_k)$ gives $O_p(\rho^{-1/\nu}N_k^\nu)$. Under $\nu < 1/(2q)$ for $q \geq 2$, we have $\min(q\nu,(1-\nu)/2) > 3\nu/2$, so the $(A_1)$ term has exponent $3\nu/2 - \min < 0 < 1 - 2\nu$, and the $E_{\mathcal{I}}$ term has exponent $\nu < 1 - 2\nu$ since $\nu < 1/3$. Thus, both are $o(N_k^{1-2\nu})$.

Combining the gradient and Hessian mismatch bounds, we obtain $(III) = o(N_k^{1-2\nu})$.

Combining the bounds $(I) = o(N_k^{1-2\nu})$ and $(III) = o(N_k^{1-2\nu})$ established above, and recalling that the middle term in the three-term decomposition is non-positive since $\tilde{\theta}_k^{(p_k)}$ maximizes $\tilde{L}_k$:

$$L_k(\theta_k^*) - L_k(\tilde{\theta}_k^{(p_k)}) \leq (I) + 0 + (III) = o(N_k^{1-2\nu}).$$

Thus, $\tilde{\theta}_k^{(p_k)}$ satisfies the approximate maximizer condition (13) with $\eta_n = o(N_k^{1-2\nu})$, which is the required rate in the framework of Shen & Wong (1994). With this condition verified, the convergence rate (12) follows by the same argument as in Tang et al. (2023) (Theorem 1). The only difference is that our estimator is an approximate rather than exact maximizer, but the slack $\eta_n = o(N_k^{1-2\nu})$ is precisely the tolerance allowed by Shen & Wong (1994) to preserve the convergence rate. This establishes (12) for $k$, completing the induction and the proof. $\square$

### D.3. Proof of Theorem 4.2

*Proof.* We follow a similar proof as shown in Tang et al. (2023) using the sieve M-theorem framework. Theorem 2 in (Tang et al., 2023) establishes asymptotic normality under six conditions: consistency (A1), four regularity conditions (A2–A3, A5–A6), and the approximate score condition (A4). Consistency (A1) is established in Theorem 4.1. The regularity conditions A2–A3, A5–A6 depend only on the model structure, and their verification under conditions (C1)–(C6) follows the same arguments as in the proof of Theorem 3 in Tang et al. (2023). It remains to verify the approximate score condition (A4).

For the infinite-dimensional component $g$, let $\mathcal{V} := \{v : v(\cdot) = \partial g_\eta(\cdot)/\partial\eta|_{\eta=0}, g_\eta \in \mathcal{G}\}$ denote the tangent space of $\mathcal{G}$. For any $v \in \mathcal{V}$, the Fréchet derivative of the log-likelihood $\ell(\beta, g; \mathcal{O})$ in direction $v$ is

$$\ell'_g(\beta, g; \mathcal{O})[v] := \frac{\partial\ell(\beta, (g + \eta v)(\cdot); \mathcal{O})}{\partial\eta}\bigg|_{\eta=0}.$$

By the chain rule, $\partial\ell/\partial\gamma_i = \ell'_g[B_{i,p}]$. Define the normalized score functions for the cumulative log-likelihood $L_k(\theta) = \sum_{j=1}^{k}\ell_j(\theta)$:

$$S_{\beta,N_k}(\theta) := \frac{1}{N_k}\nabla_\beta L_k(\theta), \quad S_{g,N_k}(\theta)[v] := \frac{1}{N_k}\sum_{j=1}^{k}\sum_{i=1}^{n_j}\ell'_g(\beta, g; \mathcal{O}_{ji})[v].$$

Condition (A4) requires that the renewable estimator $\tilde{\theta}_k = (\tilde{\beta}_k, \tilde{g}_k)$ satisfies:

$$S_{\beta,N_k}(\tilde{\theta}_k) = o_p(N_k^{-1/2}), \quad S_{g,N_k}(\tilde{\theta}_k)[\mathbf{v}^*] = o_p(N_k^{-1/2}), \tag{16}$$

where $\mathbf{v}^* \in \mathcal{V}$ is the least favorable direction. By semiparametric efficiency theory (Bickel et al., 1993), $\mathbf{v}^*$ solves the integral equation $\mathcal{I}_g\mathbf{v}^* = \mathcal{I}_{g\beta}$, where $\mathcal{I}_g$ and $\mathcal{I}_{g\beta}$ are components of the information operator. For the Cox model, this yields the explicit form (Tang et al., 2023):

$$\mathbf{v}^*(t) = \frac{\mathbb{E}\{\mathbf{1}(Y \geq t)e^{X^\top\beta_0}X\}}{\mathbb{E}\{\mathbf{1}(Y \geq t)e^{X^\top\beta_0}\}}.$$

The analysis for both components is parallel, so we focus on the $\beta$ component and verify $S_{\beta,N_k}(\tilde{\theta}_k) = o_p(N_k^{-1/2})$, or equivalently, $\nabla_\beta L_k(\tilde{\theta}_k) = o_p(N_k^{1/2})$.

We decompose the score $\nabla_\beta L_k(\tilde{\theta}_k)$ by separating the current batch from the historical batches:

$$\nabla_\beta L_k(\tilde{\theta}_k) = \nabla_\beta\ell_k(\tilde{\theta}_k) + \sum_{j=1}^{k-1}\nabla_\beta\ell_j(\tilde{\theta}_k). \tag{17}$$

Since $\tilde{\theta}_k$ maximizes the surrogate likelihood $\tilde{L}_k(\theta) = \ell_k(\theta) - \frac{1}{2}(\theta - \tilde{\theta}_{k-1})^\top\tilde{H}_{k-1}(\theta - \tilde{\theta}_{k-1}) + \text{const}$, the first-order condition $\nabla\tilde{L}_k(\tilde{\theta}_k) = 0$ yields:

$$\nabla\ell_k(\tilde{\theta}_k) = \tilde{H}_{k-1}(\tilde{\theta}_k - \tilde{\theta}_{k-1}). \tag{18}$$

In particular, for the $\beta$ component:

$$\nabla_\beta\ell_k(\tilde{\theta}_k) = [\tilde{H}_{k-1}]_{\beta\beta}(\tilde{\beta}_k - \tilde{\beta}_{k-1}) + [\tilde{H}_{k-1}]_{\beta\gamma}(\tilde{\gamma}_k - \tilde{\gamma}_{k-1}), \tag{19}$$

where $[\tilde{H}_{k-1}]_{\beta\beta} \in \mathbb{R}^{d\times d}$ and $[\tilde{H}_{k-1}]_{\beta\gamma} \in \mathbb{R}^{d\times(p_k+1)}$ are the corresponding blocks of the working Hessian.

For the historical score sum in (17), we apply Taylor expansion around $\tilde{\theta}_{k-1}$. Since $H_j = -\nabla^2\ell_j$ and using the block structure $\nabla^2\ell_j = \begin{pmatrix} \nabla_{\beta\beta}^2\ell_j & \nabla_{\beta\gamma}^2\ell_j \\ \nabla_{\gamma\beta}^2\ell_j & \nabla_{\gamma\gamma}^2\ell_j \end{pmatrix}$:

$$\sum_{j=1}^{k-1}\nabla_\beta\ell_j(\tilde{\theta}_k) = \sum_{j=1}^{k-1}\nabla_\beta\ell_j(\tilde{\theta}_{k-1}) - \sum_{j=1}^{k-1}\left([H_j(\xi)]_{\beta\beta}(\tilde{\beta}_k - \tilde{\beta}_{k-1}) + [H_j(\xi)]_{\beta\gamma}(\tilde{\gamma}_k - \tilde{\gamma}_{k-1})\right), \tag{20}$$

where for each $j$, there exists $\xi_j$ on the segment between $\tilde{\theta}_{k-1}$ and $\tilde{\theta}_k$. We write $\xi$ for notational simplicity, as the subsequent bounds hold uniformly over all such intermediate points.

Substituting (19) and (20) into (17):

$$\nabla_\beta L_k(\tilde{\theta}_k) = \sum_{j=1}^{k-1} \nabla_\beta \ell_j(\tilde{\theta}_{k-1})$$
$$+ \left([\tilde{H}_{k-1}]_{\beta\beta} - \sum_{j=1}^{k-1}[H_j(\xi)]_{\beta\beta}\right)(\tilde{\beta}_k - \tilde{\beta}_{k-1})$$
$$+ \left([\tilde{H}_{k-1}]_{\beta\gamma} - \sum_{j=1}^{k-1}[H_j(\xi)]_{\beta\gamma}\right)(\tilde{\gamma}_k - \tilde{\gamma}_{k-1}). \tag{21}$$

The term $\sum_{j=1}^{k-1} \nabla_\beta \ell_j(\tilde{\theta}_{k-1})$ equals $\nabla_\beta L_{k-1}(\tilde{\theta}_{k-1})$. Here we note that the parameters $\tilde{\theta}_{k-1}$ appearing in (21) are implicitly degree-elevated to $p_k$, i.e., $\tilde{\theta}_{k-1} = \tilde{\theta}_{k-1}^{(p_k)} := \bar{R}_{p_{k-1}\to p_k}\tilde{\theta}_{k-1}^{(p_{k-1})}$. Since the log-likelihood $\ell_j(\theta)$ depends on $\theta$ only through $(\beta, g(\cdot))$ and degree elevation preserves both components exactly, we have $\nabla_\beta \ell_j(\tilde{\theta}_{k-1}^{(p_k)}) = \nabla_\beta \ell_j(\tilde{\theta}_{k-1}^{(p_{k-1})})$ for all $j$. This invariance allows us to connect the inductive hypothesis at degree $p_{k-1}$ to the current analysis at degree $p_k$. This suggests a recursive structure, which we exploit via induction.

**Base case** ($k = 1$): The estimator $\tilde{\theta}_1$ is the sieve MLE for the first batch. By condition (C1) (ensuring $\tilde{\beta}_1$ lies in the interior of $\mathcal{B}$) and the first-order condition, $\nabla_\beta L_1(\tilde{\theta}_1) = 0 = o_p(n_1^{1/2})$.

**Inductive step:** Assume $\nabla_\beta L_{k-1}(\tilde{\theta}_{k-1}) = o_p(N_{k-1}^{1/2})$. We show this implies the same for $k$. By (21):

$$\nabla_\beta L_k(\tilde{\theta}_k) = \nabla_\beta L_{k-1}(\tilde{\theta}_{k-1}) + \Delta_k, \tag{22}$$

where the increment

$$\Delta_k = \left([\tilde{H}_{k-1}]_{\beta\beta} - \sum_{j=1}^{k-1}[H_j(\xi)]_{\beta\beta}\right)(\tilde{\beta}_k - \tilde{\beta}_{k-1})$$
$$+ \left([\tilde{H}_{k-1}]_{\beta\gamma} - \sum_{j=1}^{k-1}[H_j(\xi)]_{\beta\gamma}\right)(\tilde{\gamma}_k - \tilde{\gamma}_{k-1}).$$

We first establish bounds on the parameter increments. By Theorem 4.1:

$$\|\tilde{\theta}_k - \tilde{\theta}_{k-1}\| = O_p(N_k^{-(1-\nu)/2}). \tag{23}$$

For the $\beta$ component specifically, a sharper bound holds. From the first-order condition (18), we have

$$\tilde{\theta}_k - \tilde{\theta}_{k-1} = \tilde{H}_{k-1}^{-1}\nabla \ell_k(\tilde{\theta}_k).$$

Extracting the $\beta$ component:

$$\tilde{\beta}_k - \tilde{\beta}_{k-1} = [\tilde{H}_{k-1}^{-1}]_{\beta\beta}\nabla_\beta \ell_k(\tilde{\theta}_k) + [\tilde{H}_{k-1}^{-1}]_{\beta\gamma}\nabla_\gamma \ell_k(\tilde{\theta}_k).$$

By condition (C6), the cumulative Hessian $\tilde{H}_{k-1}$ satisfies $\lambda_{\min}(\tilde{H}_{k-1}) \geq cN_{k-1}$ for some constant $c > 0$. This implies

$$\|[\tilde{H}_{k-1}^{-1}]_{\beta\beta}\|_{\mathrm{op}} = O(N_{k-1}^{-1}), \quad \|[\tilde{H}_{k-1}^{-1}]_{\beta\gamma}\|_{\mathrm{op}} = O(N_{k-1}^{-1}).$$

For the score at $\tilde{\theta}_k$, by Taylor expansion around $\tilde{\theta}_{k-1}$ and the CLT:

$$\nabla_\beta \ell_k(\tilde{\theta}_k) = \nabla_\beta \ell_k(\tilde{\theta}_{k-1}) + O_p(n_k\|\tilde{\theta}_k - \tilde{\theta}_{k-1}\|) = O_p(\sqrt{n_k}) + o_p(\sqrt{n_k}) = O_p(\sqrt{n_k}),$$

where the remainder is $o_p(\sqrt{n_k})$ since $n_k\|\tilde{\theta}_k - \tilde{\theta}_{k-1}\| = O_p(n_k N_k^{-(1-\nu)/2}) = o_p(\sqrt{n_k})$ under $\nu < 1/2$. Similarly, $\nabla_\gamma \ell_k(\tilde{\theta}_k) = O_p(\sqrt{n_k p_k})$. Combining these bounds:

$$\|\tilde{\beta}_k - \tilde{\beta}_{k-1}\| \le \|[\tilde{H}_{k-1}^{-1}]_{\beta\beta}\|_{\mathrm{op}}\|\nabla_\beta \ell_k(\tilde{\theta}_k)\| + \|[\tilde{H}_{k-1}^{-1}]_{\beta\gamma}\|_{\mathrm{op}}\|\nabla_\gamma \ell_k(\tilde{\theta}_k)\|$$
$$= O(N_{k-1}^{-1}) \cdot O_p(\sqrt{n_k}) + O(N_{k-1}^{-1}) \cdot O_p(\sqrt{n_k p_k})$$
$$= O_p(N_k^{-1/2}) + O_p(N_k^{-(1-\nu)/2}).$$

Since $\nu < 1/2$, we have $(1-\nu)/2 > 1/4 > 0$, so both terms are $O_p(N_k^{-1/2})$. Thus:

$$\|\tilde{\beta}_k - \tilde{\beta}_{k-1}\| = O_p(N_k^{-1/2}). \tag{24}$$

We now bound $\Delta_k$.

Recall that the Hessian has the block structure $H = \begin{pmatrix} H_{\beta\beta} & H_{\beta\gamma} \\ H_{\gamma\beta} & H_{\gamma\gamma} \end{pmatrix}$. By the triangle inequality:

$$\left\|([\tilde{H}_{k-1}]_{\beta\beta}, [\tilde{H}_{k-1}]_{\beta\gamma}) - \sum_{j=1}^{k-1}([H_j(\xi)]_{\beta\beta}, [H_j(\xi)]_{\beta\gamma})\right\|_{\mathrm{op}}$$
$$\le \left\|\tilde{H}_{k-1} - \sum_{j=1}^{k-1} H_j(\theta_0)\right\|_{\mathrm{op}} + \left\|\sum_{j=1}^{k-1} H_j(\theta_0) - \sum_{j=1}^{k-1} H_j(\xi)\right\|_{\mathrm{op}}.$$

For the first term, by Theorem D.4 and the analysis in the proof of Theorem 4.1:

$$\left\|\tilde{H}_{k-1} - \sum_{j=1}^{k-1} H_j(\theta_0)\right\|_{\mathrm{op}} = O_p(N_{k-1}^{1+\nu/2-\min(q\nu,(1-\nu)/2)}) + O(\rho^{-1/\nu} N_{k-1}).$$

Since $\min(q\nu, (1-\nu)/2) > \nu/2$ under $\nu < 1/(2q)$ for $q \ge 2$, and $\rho^{-1/\nu}$ can be made arbitrarily small by choosing $\rho$ large, the first term is $O_p(N_{k-1}^{(1+\nu)/2})$.

For the second term, under condition (C2), the per-observation Hessian is Lipschitz:

$$\|H_j(\theta) - H_j(\theta')\|_{\mathrm{op}} \le C n_j \|\theta - \theta'\|$$

for some constant $C > 0$. Since $\xi$ lies on the segment between $\tilde{\theta}_{k-1}$ and $\tilde{\theta}_k$:

$$\|\theta_0 - \xi\| \le \|\theta_0 - \tilde{\theta}_{k-1}\| + \|\tilde{\theta}_{k-1} - \tilde{\theta}_k\| = O_p(N_{k-1}^{-(1-\nu)/2})$$

by Theorem 4.1 and (23). Summing over batches:

$$\left\|\sum_{j=1}^{k-1} H_j(\theta_0) - \sum_{j=1}^{k-1} H_j(\xi)\right\|_{\mathrm{op}} \le C N_{k-1} \cdot O_p(N_{k-1}^{-(1-\nu)/2}) = O_p(N_{k-1}^{(1+\nu)/2}).$$

Combining both terms, the Hessian mismatch in the $\beta$-rows satisfies:

$$\left\|([\tilde{H}_{k-1}]_{\beta\beta}, [\tilde{H}_{k-1}]_{\beta\gamma}) - \sum_{j=1}^{k-1}([H_j(\xi)]_{\beta\beta}, [H_j(\xi)]_{\beta\gamma})\right\|_{\mathrm{op}} = O_p(N_{k-1}^{(1+\nu)/2}). \tag{25}$$

For the contribution from the $\beta$ difference, using (24):

$$\left\|([\tilde{H}_{k-1}]_{\beta\beta} - \sum_{j=1}^{k-1}[H_j(\xi)]_{\beta\beta})(\tilde{\beta}_k - \tilde{\beta}_{k-1})\right\| = O_p(N_{k-1}^{(1+\nu)/2}) \cdot O_p(N_k^{-1/2}) = O_p(N_k^{\nu/2}) = o_p(N_k^{1/2}),$$

where the last step uses $\nu < 1$. For the contribution from the $\gamma$ difference, by (23) and (25):

$$\left\| ([\tilde{H}_{k-1}]_{\beta\gamma} - \sum_{j=1}^{k-1} [H_j(\xi)]_{\beta\gamma})(\tilde{\gamma}_k - \tilde{\gamma}_{k-1}) \right\| = O_p(N_{k-1}^{(1+\nu)/2}) \cdot O_p(N_k^{-(1-\nu)/2}) = O_p(N_k^{\nu}) = o_p(N_k^{1/2}),$$

where the last step uses $\nu < 1/2$, which holds under $\nu < 1/(2q)$ for $q \geq 1$.

Combining both contributions, $\Delta_k = o_p(N_k^{1/2})$. By the recursive relation (22) and the inductive hypothesis $\nabla_\beta L_{k-1}(\tilde{\theta}_{k-1}) = o_p(N_{k-1}^{1/2})$:

$$\nabla_\beta L_k(\tilde{\theta}_k) = o_p(N_{k-1}^{1/2}) + o_p(N_k^{1/2}) = o_p(N_k^{1/2}),$$

which completes the induction for the $\beta$ component.

The score condition along the least favorable direction $\mathbf{v}^*$ can be verified by an analogous argument applied to the $\gamma$ component, with the sieve projection error $\|\Pi_p \mathbf{v}^* - \mathbf{v}^*\|_{L^2} = O(p^{-q})$ contributing $o_p(N_k^{1/2})$ under $q\nu > 1/2$.

With condition (A4) verified, all conditions of Tang et al. (2023) Theorem 2 are satisfied. Therefore, the renewable estimator $\tilde{\beta}_k$ satisfies:

$$\sqrt{N_k}(\tilde{\beta}_k - \beta_0) \xrightarrow{d} \mathcal{N}(0, I^{-1}(\beta_0)),$$

where $I(\beta_0)$ is the efficient Fisher information matrix. This is identical to the asymptotic distribution of the oracle sieve MLE that has access to all $N_k$ observations, establishing oracle efficiency. $\qquad\square$

