# OpenReview forum: "Online Learning and Inference for Cox Proportional Hazards Model Using Renewable Sieve Estimation"
_ICML.cc/2026/Conference — ICML 2026 regular_

### Official Review · Reviewer_D2Tu · 2026-03-02

**Soundness:** 2
**Presentation:** 3
**Significance:** 3
**Originality:** 3
**Overall Recommendation:** 4
**Confidence:** 3

**Summary:**

This paper addresses the challenge of performing online learning and rigorous statistical inference for the Cox Proportional Hazards model in streaming data environments. The core difficulty lies in the non-decomposable nature of the Cox partial likelihood, which traditionally requires access to full historical data to construct global risk sets, thereby creating bottlenecks for memory efficiency and data privacy. To overcome this, the authors propose COLSA (Collaborative Operation of Linked Survival Analysis), a framework that utilizes renewable sieve estimation.

The method approximates the infinite-dimensional log-baseline hazard function using Bernstein basis polynomials. A key innovation is the algorithm's ability to dynamically adapt model complexity; as the data stream grows, the polynomial degree increases via a two-stage projection mechanism (Progressive Activation and Pre-estimation) that maps historical summary statistics to higher-dimensional spaces without accessing raw data. Theoretically, the authors prove that COLSA achieves asymptotic consistency and normality, matching the statistical efficiency of a centralized "Oracle" estimator. Empirically, the approach is validated through simulations and an application to the SRTR kidney transplant database, demonstrating superior bias reduction and inference accuracy compared to existing online approximations.

**Compliance With Llm Reviewing Policy:**

Affirmed.

**Final Justification:**

The authors have adequately addressed my main concerns in the rebuttal. I have increased my score accordingly

**Key Questions For Authors:**

1. Discrepancy in Real-World Results: In Table 2, the COLSA estimator for the "Hispanic" recipient coefficient ($Z = -2.22$) deviates noticeably from the Oracle estimator ($Z = -1.58$) in terms of Z-statistics, despite the claim of asymptotic equivalence to the Oracle. Could you explain the theoretical or numerical source of this specific divergence? Does this suggest potential instability in the sieve approximation when dealing with sparse subgroups or specific covariate distributions? A clarification here is crucial for assessing the reliability of the method in diverse real-world scenarios.

2. Computational Bottleneck & Optimization: The runtime analysis indicates that COLSA is significantly slower (28s for $K=200$) compared to existing online SGD methods (1.6s). The paper attributes this to numerical integration. Have you explored or analyzed the trade-off of using faster approximation techniques (e.g., quadrature methods or discrete approximations) for the integral term? If the computational cost can be reduced without significantly violating the "Oracle" efficiency bounds, it would greatly enhance the practical applicability of the framework.

3. Hyperparameter Sensitivity in Streaming: The dynamic degree expansion relies on hyperparameters $\nu$ (growth rate) and $C$ (scaling constant). In a true online streaming setting where future data distributions are unknown, performing grid search is often infeasible. Can you provide a sensitivity analysis or a heuristic guideline for selecting these parameters robustly in a one-pass setting? Understanding the method's stability with suboptimal hyperparameters would strengthen the claim of its utility in automated monitoring systems. To be honest, I don't completely understand that how the authors choose $\nu$ and $C$.

4. Generalization to Other Datasets: The empirical validation is currently limited to the SRTR dataset. Given the complexity of the proposed two-stage update mechanism, demonstrating robustness across different data structures is important. Do you have results on other standard survival analysis datasets that confirm the consistency of the findings, particularly regarding the variance reduction capabilities compared to mini-batch SGD?

**Limitations:**

The authors should explicitly discuss the computational trade-offs of COLSA. While theoretically efficient, the method is significantly slower (approx. 26x in experiments) than standard online SGD due to numerical integration; acknowledging this runtime overhead and defining the specific scenarios where this latency is acceptable (e.g., daily updates vs. millisecond trading) would strengthen the paper. Additionally, the authors should address the hyperparameter sensitivity in true online settings. Since grid search is often infeasible in one-pass streaming, discussing the robustness of the method to suboptimal choices of the growth rate ($\nu$) and scaling constant ($C$), or the lack of an automated selection mechanism, is a necessary limitation to disclose. Finally, the assumption of a fixed covariate dimension ($d$) should be highlighted as a constraint that currently prevents the method's application to high-dimensional data (e.g., genomics), which is common in survival analysis.

**Strengths And Weaknesses:**

1. Strengths

1.1. Theoretical Soundness: The paper is theoretically strong. The authors provide rigorous proofs (Theorems 4.1 and 4.2) showing that their estimator achieves asymptotic normality and efficiency equivalent to the full-data partial likelihood estimator. This is a significant step up from many heuristic online approximations.

1.2. Presentation: The paper is generally well-written and structured. The transition from the standard Cox model to the sieve approximation is logical.

1.3. Significance: The work addresses a critical bottleneck in healthcare and industrial reliability: performing rigorous survival analysis on streaming data without violating privacy or memory constraints. By eliminating the need for global risk sets, COLSA theoretically enables privacy-preserving, distributed inference that matches centralized "Oracle" performance. This has high potential impact for federated learning applications in medical research where data cannot be centralized.

1.4. Originality: The paper demonstrates high originality by innovatively combining renewable estimation with sieve maximum likelihood estimation to solve the non-decomposable loss problem in Cox models. The specific contribution of the "two-stage" update mechanism—utilizing the recursive structure of Bernstein polynomials to dynamically expand the parameter space (degree $p$) without accessing historical data—is a novel technical advancement. The introduction of "shadow" sufficient statistics to prepare for future dimension expansion is a creative solution to the stability issues often found in online learning.

2. Weaknesses:

2.1. Empirical Soundness: This is the primary weakness. The experimental validation is somewhat thin for a top-tier conference. The method is evaluated on only one real-world dataset (SRTR), whereas standard practice typically requires 2-3 diverse datasets to demonstrate generalization. Furthermore, there is a concerning discrepancy in the real-world application (Table 2) where in Ref: white the COLSA estimator for the "Hispanic" coefficient (Z-statistics=$-1.58$) differs notably in magnitude from the Oracle (Z-statistics=$-2.22$), suggesting potential difference between the asymptotic distribution of Oracle estimator and COLSA estimator. The simulation settings are also somewhat limited (only mixture Weibull distributions).

2.2. Presentation: The notation regarding the dynamic degree updates (switching between active degree $p_k$ and pre-estimation degree $\bar{p}_k$) is dense and requires significant effort to parse. The algorithmic description (Algorithm 1) helps, but the complexity of the matrix projections could be clarified with a schematic diagram illustrating the information flow between batches.

2.3. Significance: The practical significance is slightly tempered by the computational overhead compared to simpler SGD methods, which might limit its use in ultra-high-frequency trading or monitoring systems.

---

> ### Author Rebuttal · Authors · 2026-03-30
>
> We sincerely appreciate the reviewer's thorough review and insightful feedback. For weakness 2.2 Presentation, we created a diagram [(Link)](https://doi.org/10.6084/m9.figshare.31888510). Below, we address the remaining points.
>
> **Weakness 2.1/ Q1** We thank the reviewer for noting the discrepancy in the "Hispanic" Z-scores. This is mainly due to the very low event rate (~1.6\%) in this subgroup as noted in Section 5.2. It is well known in survival analysis that the Cox model estimates may become numerically unstable in the presence of rare events, particularly in finite or small samples. The finite-sample instability does not contradict our large-sample asymptotic results. Even in this challenging setting, COLSA remains the closest to the Oracle. The results from other methods deviate from COLSA by varying levels of discrepancy:
>
> | Method | FN | FP |
> |---|---|---|
> |Oracle|0|0|
> |COLSA|1|0|
> |Meta|3|2|
> |Online|2|2|
> |SGD|7|3|
>
> A standard solution in the classical survival analysis is to merge sparse categories. For example,  combining the "Hispanic" with "Other" (event rate ~0.6%) categories to boost the event rate modestly, the numerical stability gets greatly improved, yielding Z-score of −3.60 (Oracle) and −2.92 (COLSA), implying that COLSA reaches the same inferential conclusion as the oracle.  This analysis leads to a suggestion of practical relevance: to be cautious with categorical variables with low event rates, and combining some categories of low event rates is recommended as part of sensitivity analysis. We will add this important point to the Discussion.
>
> **Weakness 2.3/Q2**  While SGD is faster, our analysis (Table 1, Figure 1) shows that it produces biased estimates and invalid standard errors for inference. In SRTR application, this resulted in 10 inferential errors (7 FN, 3 FP), whereas COLSA had one marginal discrepancy. Thus, the key trade-off is valid inference vs. computation speed. COLSA is designed for applications where reliable uncertainty quantification is required. To further improve computation efficiency, we propose:
>
> * Integration over unique event times: Inspired by your suggestion, we utilize the discrete nature of event times. Survival data often contain tied event times (e.g., 500 unique times among 48,766 SRTR patients). Performing integration only over unique event times rather than for every single observation avoids redundant calculations and can greatly reduce runtime without accuracy loss.
>
> * Software/Hardware Optimization: We currently use the default Gaussian quadrature in R. Developing a C++/Rcpp implementation can reduce processing time by an order of magnitude. Combined with hardware improvements, this makes COLSA feasible for many real-time and large-scale applications.
>
> These will be added as "Integration over unique event times" strategy to Discussion of the revised manuscript.
>
> **Q3**  We agree this was not clearly explained. In practice, COLSA requires minimal tuning. Neither $\nu$ nor $\rho$ requires tuning. $\nu$ is determined by the smoothness of $g_0$. Theorems 4.1-4.2 give $1/(2(q+1)) < \nu < 1/(2q)$. For $q=2$, $\nu=0.2$ is recommended as the default.  $\rho$needs only to be $ > 1$ (Theorem C.4: error $O(\rho^{-1/\nu})$. $\rho=2$ gives < 5% error). Empirical results confirm $\rho \in (1.5, 2.0, 2.5, 3.0)$ all produce almost identical results [(Link)](https://doi.org/10.6084/m9.figshare.31888576). $C$ is the only data-driven parameter, obtained as $C = p_1^*/n_1^{\nu}$ where $p_1$ is selected via AIC on the **first** batch. After initialization, the chosen $C$ is fixed for batch 2 and beyond, where the sieve degree $p_k$ is estimated from $\lceil C N_k^{\nu} \rceil$.
>
> **Q4** We agree that demonstrating robustness across diverse data is critical. To address this, we added a second validation using the TCGA Pan-Cancer dataset, examining the relationship between gene expression and survival across cancer types. A ground-truth set of $d = 23$ genes was selected from 46,219 candidates, passing a Bonferroni-adjusted $P$-Value in pooled univariate Cox regression and then were further refined via LASSO Cox regression with 5-fold cross-validation.
>
> Data were streamed across $K = 18$ batches for different cancer types with $n$ ranging from $1{,}073$ for BRCA to $85$ for MESO to simulate an online setting, and a multivariate Cox model was used to model the relationship between survival and selected genes. COLSA closely matched the offline Oracle model, recovering 22 of 23 genes, whereas Meta-analysis missed 5, and the Online baseline and SGD disagreed on 12 and 11 genes, respectively.
>
> Full results [(Link)](https://doi.org/10.6084/m9.figshare.31888549) and analysis code will be provided in the paper and supplementary material.
>
> **Fixed covariate dimension**:  The current paper assumes fixed $d$. Extending it to high-dimensional settings via debiased regularization is a critical future direction (see second paragraph of our response to reviewer dFi9 for details).

---

> > ### Author Rebuttal · Reviewer_D2Tu · 2026-03-31
> >
> > The authors have adequately addressed my main concerns in the rebuttal. I have increased my score accordingly

---

> > > ### Author Response · Authors · 2026-04-03
> > >
> > > Thank you for your review and all the insightful suggestions. We are pleased that our rebuttal addressed your primary concerns and sincerely appreciate your decision to raise the score.

---

### Official Review · Reviewer_dFi9 · 2026-03-10

**Soundness:** 4
**Presentation:** 3
**Significance:** 3
**Originality:** 4
**Overall Recommendation:** 5
**Confidence:** 4

**Summary:**

This study proposes the COLSA online learning framework. It uses dynamically expanding polynomials to approximate the Cox baseline hazard. The method maintains sufficient statistics for a higher-order basis and employs data-driven basis projection to adaptively scale model complexity to the effective sample size. It achieves Oracle-level inference without storing historical data.

**Compliance With Llm Reviewing Policy:**

Affirmed.

**Final Justification:**

I accept the authors' explanation, and I maintain my already positive score.

**Key Questions For Authors:**

1) This article uses a "Shadow Hessian" and the Moore-Penrose pseudoinverse to convert polynomial degrees. Will this cause the Hessian matrix to lose positive definiteness in real streaming data? If so, how should this be handled?
2) This article outlines the model complexity. When data streams reach massive scales, will the storage overhead and inversion computation of the Hessian matrix break the resource limits of edge nodes? Please discuss this in detail. Should a "pruning" mechanism be introduced to fix the upper limit of model complexity?
3) How does the method ensure stability if extreme values occur in the data? What preprocessing is needed to limit the data characteristics applicable to this method?

**Limitations:**

This is mentioned in the future work section. However, the discussion and analysis on small sample sizes and high-dimensional feature scenarios could be more detailed.

**Strengths And Weaknesses:**

Strengths: The study updates survival models online while strictly protecting privacy. It avoids global risk sets. It retains statistical testing capabilities. It addresses the pain point that the Cox model cannot be decomposed. The step-up and step-down matrix projection algorithm (Algorithm 1） is detailed. It provides a hyperparameter selection strategy. It proves the consistency and Oracle-level theoretical properties of the renewable sieve estimator.
	Weaknesses: 1. High-dimensional feature scenarios are important in practice. This article lacks a discussion on the method's applicability in this context. 2. Detailed discussions are needed.

---

> ### Author Rebuttal · Authors · 2026-03-30
>
> We sincerely thank the reviewer for the thorough review and constructive feedback. Below, we provide the point-by-point response.
>
> **Weaknesses 1/2:**  We thank the reviewer for raising this excellent point. Regularization methods, such as $L_1$ penalty, can be applied to COLSA in high-dimensional streaming settings to reduce dimensionality. For the online LASSO method, the fundamental challenge remains in the necessity of debiasing for valid inference, making this a non-trivial extension for the semi-parametric Cox model.
>
> Existing literature has begun to address these challenges in Generalized Linear Models by integrating debiasing terms with online error-correction in [1] while recent methods, such as the Adaptive Online Debiased Lasso [2], offer ways to significantly reduce associated storage costs. The two-stage sieve degree elevation utilized in COLSA could be integrated into these debiased frameworks to facilitate dynamic estimation within the nuisance parameter space.
>
> We have added a discussion of this extension in the new "Discussion and Future Work" section.
>
> [1] Online inference in high-dimensional generalized linear models with streaming data. Electron. J. Stat.  2023
>
> [2] Adaptive debiased lasso in high-dimensional generalized linear models with streaming data. JASA, 2026.
>
> **Question 1**
>  We thank the reviewer for raising the concern. The degree elevation $R_{p \to p'}^\top \tilde{H} R_{p \to p'}$ preserves positive definiteness because $R_{p \to p'}$ has full column rank (Remark 3.1). The pseudoinverse projection (Algorithm 1, line 21)  expands the shadow Hessian to a higher-dimensional space, producing a semi-definite intermediate. This design allows the shadow Hessian to serve as a warm start, and positive definiteness is restored once the new batch's Hessian $H_k$ is incorporated. Theorem~C.4 bounds the overall approximation error at $O(\rho^{-1/\nu})$ under conditions (C1) - (C6). No numerical issues were observed in either the simulations or the real-data analysis.
>
> **Question 2**
> We thank the reviewer for raising this important practical concern and the suggestion on setting an upper limit. In the standard low-dimensional covariate setting, which is the focus of this paper, COLSA is theoretically impossible to incur a massive Hessian matrix. A key advantage of COLSA is that its storage overhead is entirely independent of the number of batches $K$. The memory and computational bottleneck is dictated solely by the dimension of the Hessian matrix, $\mathcal{O}(d +\bar{p}\_k)$, where $d$ is the covariate dimension and $p_k$ is the sieve degree. Specifically, $\bar{p}\_k = \rho C N_k^\nu$ grows at a very slow, sublinear rate (default $\nu = 0.2$). This well controls the dimension of Hessian such that at $N=10^3$ samples, $\bar{p}\_k \approx 4$, and even at a massive scale of $N=10^6$ samples, $\bar{p}\_k$ only grows to $\approx 16$. In our extensive experiments, the full Hessian matrix never exceeded a $26 \times 26$ dimension.
>
> Storing and inverting a matrix of this size should operate comfortably within the resource limits of standard machines.
>
> We acknowledge that if the dataset has a massive number of features, like in the high-dimensional case discussed above, the Hessian would indeed risk breaking resource limits. In such contexts, the standard COLSA algorithm would need to be adapted. As we note in the above reply, this can be resolved by coupling COLSA with approximated online debiased Lasso techniques, which decompose the Hessian into low-dimensional summary statistics and bypass direct matrix inversion entirely, reducing the storage requirement.
>
> **Question 3**
> We thank the reviewer for raising this important practical concern regarding extreme values. If the data stream contains severe outliers, the standard unweighted likelihood can experience instability. To ensure robustness against such anomalies, COLSA's incremental score equation can be naturally extended by adopting the robust likelihood-downweighting framework [3] that downweights the influence of observations with extreme martingale residuals.
>
> Regarding data preprocessing, we recommend normalizing continuous variables and clipping at a reasonable threshold, such as having Z score above  3 standard deviations, to prevent extreme values from causing numerical overflow or destabilizing Hessian updates. We will add this in the discussion section to explicitly outline these recommendations.
>
> [3] weighted partial likelihood estimators for the cox model. JASA 1993.

---

> > ### Author Rebuttal · Reviewer_dFi9 · 2026-04-03
> >
> > I accept the authors' explanation, and I maintain my already positive score.

---

> > > ### Author Response · Authors · 2026-04-03
> > >
> > > Thank you for your time in providing a thorough review and for acknowledging our rebuttal. We truly appreciate your positive assessment.

---

### Official Review · Reviewer_2G2B · 2026-03-15

**Soundness:** 3
**Presentation:** 3
**Significance:** 3
**Originality:** 3
**Overall Recommendation:** 5
**Confidence:** 3

**Summary:**

This paper focuses on an online learning framework for survival analysis, and propose a new method for estimating the parameters of the Cox proportional hazards model. Their approach uses Bernstein basis polynomials to approximate the log-baseline hazard. They show that under certain regularity conditions their estimator is consistent and asymptotically normal. They then assess the performance of their model---against other proposed models---via simulations. They show that their model outperforms existing models.

**Compliance With Llm Reviewing Policy:**

Affirmed.

**Final Justification:**

My final recommendation is to accept the paper. The rebuttal addressed my main concerns, and I had initially recommended to accept the paper.

**Key Questions For Authors:**

(1)	Why is the incremental score equation the thing we want to be after?
(2)   You should consider defining the truncation time before you reference it in section 3.2.
(3) Could you provide more simulations to assess how your method compares against existing methods? A semi-synthetic experiment would further improve this paper.

**Limitations:**

yes

**Strengths And Weaknesses:**

The paper does a good job at explaining the limitations of existing methods and how their method addresses those limitations. Nonetheless, their discussion about the computational and memory tradeoffs with existing methods is only found in the Experiments section. It would be nice to see this discussion earlier in the paper. I would also suggest that they have some discussion on why the incremental score equation is the right mathematical object of interest.

I appreciate the author's theoretical results and their experiments. Nonetheless, I would've liked to see a further discussion on the regularity conditions and what they would imply for practitioners. I know there is a space constraint, but maybe the authors could move the formal presentation of the algorithm to the appendix, I think the authors do a good enough job at describing the algorithm throughout the paper such that the formally stated algorithm is not fully necessary.


Some minor comments:
- Equation (2), consider moving $(\theta_{k-1} - \theta_k)$ to before $\H_{k-1}$. Since $H_k(\theta)$ is function of $\theta$ this ordering confuses the reader (notice that I'm not including the tilde in the notation from Eq. 2).

---

> ### Author Rebuttal · Authors · 2026-03-30
>
> We appreciate these great suggestions for improving the flow and readability of the manuscript. We agree with you and will move Algorithm 1 to the Appendix and replace it with a simplified pseudocode. We will also utilize the space to include a dedicated session for memory and computation.
> > further discussion on the regularity conditions
>
> We thank the reviewer for the helpful suggestion. We will include a brief discussion in Section 4.1 summarizing their practical implications to improve readability.
>
> Conditions (C1)–(C2) ensure that the parameter space is well-behaved and that the baseline hazard function is sufficiently smooth to be approximated accurately.
>
> Conditions (C3)–(C4) are standard data quality requirements, implying that the covariates exhibit sufficient variation and that the censoring rate is not excessive.
>
> Finally, Conditions (C5)–(C6) are non-degeneracy assumptions that guarantee identifiability and stable estimation.
>
> As suggested by the reviewer, we will move the full version Algorithm 1 to the appendix and retain a short pseudocode in the main text to improve readability, so that the paper remains self-contained.
>
> >Equation (2), consider moving...
>
> We thank the reviewer for the helpful suggestion. We retain the current ordering to ensure dimensional consistency between the gradient and Hessian terms in the update. To improve clarity, especially at the first appearance of the incremental score equation, we will expand  $\tilde{H}\_{k-1}$ as $ \sum_{j=1}^{k-1} \tilde{H_j}(\theta_j)$ in equation (2) rather than introducing the shorthand immediately.  We will then define $\tilde{H}_{k-1}$ right after this equation to use it throughout the remainder of the section for simplifying notations.
>
> >(1) Why is the incremental score equation the thing we want to be after?
>
> Equation (2) serves as a surrogate for the full-data global score equation that would be obtained if all data batches were available at once. Specifically, the first term $U_k({\theta_k})$ is evaluated on the current data batch, while the second term aggregates information from previous batches. The second term is formulated as an optimally weighted difference between the current parameter ${\theta_k}$  and the previous estimate $\theta_{k-1}$. It effectively shrinks the new update toward the historical baseline. The magnitude of this shrinkage is dynamically governed by the cumulative Hessian matrix, $\sum_{j=1}^{k-1} H_j(\theta_j)$, which mathematically scales the penalty based on the statistical quality and curvature of the previous estimation.
>
>
> > (2) You should consider defining the truncation time.
>
> Agreed. To improve the readability, we will move the formal definition of truncation time, which is the maximum follow-up time to a general notation paragraph in Section 3.1 Problem Formulation.
>
> > (3) Could you provide more simulations... A semi-synthetic experiment would further improve this paper.
>
> We thank the reviewer for this excellent suggestion on adding additional validation experiments. To improve on this, we have added a new analysis using the TCGA Pan-Cancer genomic dataset to evaluate the relationship between gene expression and overall survival. To benchmark the estimation and inference capabilities of the various methods, we established a ground-truth set of $d = 23$ genes from 46,219 candidates. We first screened all protein-coding genes via pooled univariate Cox regression, retaining those that passed a Bonferroni-adjusted $P$-value threshold ($10^{-4}$), and subsequently applied LASSO Cox regression with 5-fold cross-validation.
>
> To evaluate performance in online learning setting, data were sequentially streamed across $K = 18$ batches for different cancer types with batch sizes ranging from $n = 1{,}073$ for BRCA and $n = 85$ for MESO. Multivariate Cox regression was used to model the relationship between survival and all 23 genes.
>
> COLSA demonstrated high consistency with the Oracle, correctly recovering 22 of the 23 gene candidates. In comparison, Meta-analysis missed 5 Oracle-significant genes, while the 'Online' baseline and SGD disagreed on 12 and 11 decisions, respectively. We will provide the full results [(Link)](https://doi.org/10.6084/m9.figshare.31888549) in the paper and the analysis code in the supplementary material.
>
> Furthermore, to provide a complete picture of how COLSA compares to existing methods under higher sparsity and small batch size, we respectfully direct the reviewer to the expanded simulation studies in Appendix B (will be more prominently referenced in Section 5.1. These simulations specifically assess the robustness of COLSA performance under some data sparsity by restricting local batch sizes to $n_k=100$. While other methods suffer from computational instability and bias due to a lack of sufficient local information, COLSA maintains nearly identical performance to the centralized Oracle.

---

> > ### Author Rebuttal · Reviewer_2G2B · 2026-04-03
> >
> > I think the authors adequately answered all of my questions/concerns.

---

> > > ### Author Response · Authors · 2026-04-05
> > >
> > > Thank you for your thorough review and for confirming that our clarifications fully resolved your concerns. We are deeply grateful for your time, valuable suggestions, and strong support of our paper.

---

### Official Review · Reviewer_ooNF · 2026-03-19

**Soundness:** 3
**Presentation:** 3
**Significance:** 3
**Originality:** 2
**Overall Recommendation:** 4
**Confidence:** 3

**Summary:**

This paper studies online learning and inference for the Cox proportional hazards model under streaming setting. Asthe standard partial likelihood is problematic for online updating, the paper proposes COLSA, which replaces the partial likelihood with a full-likelihood formulation using a sieve approximation of the log-baseline hazard via Bernstein polynomials, and then performs renewable second-order updates. A key technical component is a dynamic basis-expansion mechanism that allows the sieve dimension to grow with sample size. The paper establishes consistency and asymptotic normality results. Empirically, the method has closed efficiency to oracle and outperforms other online benckmarks.

**Compliance With Llm Reviewing Policy:**

Affirmed.

**Final Justification:**

My final recommendation remains Weak Accept. The paper addresses online learning and inference for the Cox model under streaming settings, and has clear strengths in technical soundness, theoretical development, and clarity. The reformulation via a sieve full-likelihood and the dynamic basis-expansion mechanism are interesting, and the consistency and asymptotic normality results are strong.

The rebuttal was helpful in clarifying the novelty relative to prior renewable estimation and one-pass nonparametric estimation work, and I appreciate the additional real-data experiment and the responses to my questions on memory, federated deployment, and hyperparameter sensitivity. These points improved my understanding of the paper. Overall, I still find the novelty somewhat incremental relative to prior workr. So I view the paper as technically solid and worthwhile,  with a weak accept recommendation.

**Key Questions For Authors:**

**Question 1**: COLSA's storage is  $O(d+\bar{p})^2 $ , which may become substantial as sieve size grows. The abstract’s “only requiring constant memory” wording seems too strong, since the retained state grows with  the pre-estimation degree $\bar{p}\propto N^\rho $. Could the authors clarify this claim and better explain the practical memory requirements of the method?

**Question 2**: The paper highlights potential privacy benefits in distributed or federated settings. Could the authors elaborate on how the proposed approach would be deployed in such settings in practice? A more concrete discussion of the workflow and privacy-related advantages would be very interesting.

**Question 3**: How sensitive is the empirical performance of the method to the choice of hyperparameters? Some discussion or ablation on this point would help assess the robustness of the approach.

**Question 4**: Could the authors provide more intuition for Condition 5 and explain what role it plays in the theoretical analysis? A brief interpretation would make the assumptions easier to understand.

**Limitations:**

yes

**Strengths And Weaknesses:**

**Strengths**: This paper addresses an important challenge in survival analysis under streaming settings, namely the non-decomposability of the Cox partial likelihood. The reformulation based on a sieve full likelihood is interesting and well motivated. The theoretical results are strong, and the empirical findings are also encouraging.


**Weakness 1**: Regarding technical novelty, the proposed method appears to build substantially on [1] and [2]. It would strengthen the paper if the authors could more clearly articulate the precise methodological advances beyond these prior works. In particular, what are the key differences between the proposed two-stage update mechanism and the approach in [1]? What specific challenges prevent the method in [1] from being directly applied to the present setting? A clearer discussion of these points would make the contribution more convincing.

**Weakness 2**: The empirical evaluation feels somewhat limited for an ICML submission. In particular, the paper includes only one real-data application. If time and space permit, it would be valuable to include additional real-world datasets to further demonstrate the practical effectiveness and robustness of the proposed method.


[1] Luo, L., & Song, P. X. K. (2020). Renewable estimation and incremental inference in generalized linear models with streaming data sets. Journal of the Royal Statistical Society Series B: Statistical Methodology, 82(1), 69-97.

[2] Quan, M., & Lin, Z. (2024). Optimal one-pass nonparametric estimation under memory constraint. Journal of the American Statistical Association, 119(545), 285-296.

---

> ### Author Rebuttal · Authors · 2026-03-30
>
> We sincerely thank the reviewer for the insightful comments and feedback. Below, we provide the point-by-point responses.
>
> **Weakness 1:**  We thank the reviewer for the opportunity to clarify our novelty. While [1] and [2] are foundational, they do not address the central theme of our work, which is establishing valid statistical inference for semi-parametric models in an online learning setting. [1] is designed for fully parametric models with fixed parameter dimensions. It cannot be directly applied to the semi-parametric Cox model, given that the parametric and nonparametric components exhibit different convergence rates, requiring rigorous methodology and theory developments. [2] introduces a pre-estimation strategy for point estimation but does not address the critical question of statistical inference.
>
> COLSA explicitly resolves these gaps by establishing asymptotic normality, enabling oracle-level inference not achieved in prior works. We will revise the Introduction to articulate these distinctions.
>
> **Weakness 2:** We agree with the reviewer on adding more real-world examples. We added TCGA Pan-Cancer genomic data analysis. It explores the relationship between gene expression and survival time across distinct cancer types. To benchmark the estimation and inference capabilities, we established a ground-truth set of $d = 23$ genes from an 46,219 candidates.  Specifically, we first screened all protein-coding genes via pooled univariate Cox regression, retaining those that passed a Bonferroni-adjusted $P$-value, and subsequently applied LASSO Cox regression with cross-validation.
>
> To evaluate performance in online learning setting, data were streamed across $K = 18$ batches, representing different cancer types with batch sizes, from BRCA ($n = 1{,}073$) to MESO ($n = 85$). Multivariate Cox regression was then applied to the selected 23 genes. COLSA demonstrated high consistency with Oracle, correctly recovering 22 of the 23 gene candidates. In comparison, Meta missed 5 Oracle-significant genes, while the Online and SGD disagreed on 12 and 11 decisions [(Link)](https://doi.org/10.6084/m9.figshare.31888549). This additional study underscores the practical effectiveness and robustness of COLSA in heterogeneous settings.
>
> **Question 1** We thank the reviewer's comment and will revise the "constant memory" phrasing for mathematical rigor. The storage of historical information is independent of $K$ and is much smaller than the cumulative $N$. While $p_k = C N_k^{\nu}$ grows with batch size, $\nu=0.2$ is slow ($p\approx 4$ at $n=10^3$, $p\approx 16$ at $n=10^6$). In our experiments, the Hessian never exceeds $26 \times 26$.
>
> **Question 2** Unlike traditional distributed survival analysis, which requires sharing event times with a semi-trusted data center to build global risk sets, COLSA uses a one-pass, sequential summary statistics-passing architecture.
>  - Local Processing: Site $k$ receives summary statistics from Site $k-1$.
> -  Update: Site $k$ updates these statistics using local data.
> -  Forwarding: Site $k$ passes the updated statistics to Site $k+1$ and clears local memory.
>
> Under this workflow, raw patient data never leaves the local network. Crucially, the COLSA estimator achieves a convergence rate based on the total combined sample size, which is statistically superior to the local-rate convergence of standard divide-and-conquer estimators. A dedicated federated learning platform can enable sites to independently run COLSA and securely forward summary statistics. We will discuss this in the revised manuscript.
>
> **Question 3** Simulation testing varying the $C$ parameter selection criterion (AIC vs. BIC) and $\rho \in {(1.5, 2.0, 2.5, 3.0)}$ is added here [(Link)](https://doi.org/10.6084/m9.figshare.31888576).  It shows that the results are essentially unchanged (differences at the 4th decimal place). The remaining parameter $\nu = 0.2$ is set by Theorems 4.1-4.2. Overall, COLSA has minimal hyperparameter sensitivity in practice and only requires mild data-driven calibration on the initial data batch for $C$.
>
> **Question4** Considering a one-dimensional projection $Z = u^{T} X$ and the condition is essentially $Var(Z) \geq \eta E(Z^2)$. Using the identity $Var(Z) = E(Z^2) - E(Z)^2$, this can be rewritten as $E(Z)^2/E(Z^2)\leq 1-\eta$. This implies that the ratio between the squared first moment and the second moment is constrained, and the $Var(Z)$ is non-zero. The conditional version in (C5) enforces this after conditioning on $(U,V)$ (i.e., patients with the same baseline risk and survival outcome). In other words, individuals in the same subgroup are not nearly identical in all covariates.
>
> This non-degeneracy condition is rather mild and prevents the covariates from being almost deterministic given $(U,V)$, which would make it difficult to distinguish the effect of different components of $X$. We will clarify that (C5) is for the identifiability of $\beta$ in the revised manuscript.

---

> > ### Author Rebuttal · Reviewer_ooNF · 2026-04-01
> >
> > Thanks to the authors for their response and clarifications. I will keep my score unchanged.

---

> > > ### Author Response · Authors · 2026-04-03
> > >
> > > Thank you for reviewing our rebuttal and for confirming that your concerns are fully resolved. We are grateful for your valuable feedback and your time.

---

### Official Review · Reviewer_wqwf · 2026-03-22

**Soundness:** 2
**Presentation:** 3
**Significance:** 3
**Originality:** 3
**Overall Recommendation:** 4
**Confidence:** 4

**Summary:**

This paper introduces a novel method called COLSA (Cox Online Learning with Sieve Approximation). It addresses the challenge of performing survival analysis in a streaming or federated setting where data arrives in batches and cannot be stored in full. The paper presents a framework for online estimation and inference in the Cox proportional hazards model without storing historical data or constructing global risk sets. This paper makes a contribution to the field of online survival analysis.

**Compliance With Llm Reviewing Policy:**

Affirmed.

**Final Justification:**

The authors have satisfactorily addressed my primary concerns in the rebuttal, and I have raised my score accordingly.

**Key Questions For Authors:**

Here are suggestions the author(s) should consider:
1. Computational Trade-offs Compared to Simpler/Classical Methods

Since this paper uses a second-order update that aggregates gradients and Hessians to ensure statistical efficiency, it inherently requires more time and memory. The authors should provide a more detailed comparison with the classical Meta method, as from Figure 1 and Table 2, the performance of these two methods appears similar. However, the computational costs differ substantially: for K=200, Meta requires only 1.585 seconds, whereas COLSA requires 28.36 seconds. A discussion on when the additional computational expense of COLSA is justified—particularly in terms of statistical efficiency gains or scenarios where Meta may fail—would help readers understand the practical trade-offs.

2. Handling High-Dimensional Covariates

The current framework assumes a fixed number of covariates d, but in high-dimensional streaming settings, the authors should discuss how COLSA could be extended to handle high-dimensional covariates.

3. Implementation Complexity

The algorithm involves multiple projections, degree elevation, and shadow Hessians, which may be challenging to implement and tune. Providing a simplified pseudocode, a reference implementation, or practical guidelines for practitioners would enhance the accessibility and reproducibility of the method.

4. Hyperparameter Sensitivity

The method relies on several hyperparameters (C,ν,ρ) with specific recommended values. While AIC-based selection for C is data-driven, the sensitivity of COLSA to these hyperparameter choices in practice is not fully explored. The authors should include a sensitivity analysis or robustness study across different hyperparameter settings to demonstrate the stability of the method and guide users in real-world applications.

5. Privacy Guarantees

While the method transmits only summary statistics (gradients and Hessians) rather than raw data, it does not provide formal privacy guarantees. The authors should mention this limitation and propose future extensions to incorporate formal privacy mechanisms, such as differential privacy.

**Limitations:**

yes

**Strengths And Weaknesses:**

Strengths: The paper addresses a well-known limitation of the Cox model in distributed/streaming settings: the non-decomposable partial likelihood. The the submission clearly written and well structured.

---

> ### Author Rebuttal · Authors · 2026-03-30
>
> We sincerely thank the reviewer for the thorough review and for recognizing the significance of our work. Below, we respond to your insightful suggestions:
>
> **1. Computational Trade-offs** We recognize Meta's speed advantage, but it comes at the expense of valid statistical inference. COLSA was specifically developed to prioritize reliable uncertainty quantification, which is strictly required in clinical or monitoring settings.  As demonstrated in Fig 1 and 2 (Appendix),  Meta exhibits substantially higher bias (e.g., for K=200, Meta's absolute bias is 2.16 times as large as COLSA’s), and it also suffers from compromised coverage probabilities (73.03\%, well below the 95\% nominal level). In the SRTR example, this causes Meta to make 2 false positives and 3 false negatives out of 20 coefficients, making it unsuitable for settings where valid inference is essential.
>
> The additional computational cost of COLSA is justified by its ability to achieve near-Oracle performance in estimation and inference.
>
> **2. Handling High-Dimensional Covariates**: We thank the reviewer for highlighting this critical future direction. $L_1$ regularization can be applied to COLSA in high-dimensional streaming settings. The necessary debiasing for valid LASSO inference makes this a non-trivial extension. Existing literature tackles these challenges in Generalized Linear Models by integrating debiasing terms with online error-correction in [1], while recent methods like the Adaptive Online Debiased Lasso [2] offer ways to significantly reduce associated storage costs. The two-stage sieve degree elevation utilized in COLSA could be integrated into these debiased frameworks to facilitate dynamic estimation within the nuisance parameter space.
>
> We will discuss this extension in the new "Discussion and Future Work" section.
>
> [1] Online inference in high-dimensional generalized linear models with streaming data. Electron. J. Stat.  2023
>
> [2] Adaptive debiased lasso in high-dimensional generalized linear models with streaming data. JASA, 2026.
>
> **3. Implementation Complexity**: We agree with the reviewer. To make the algorithm more accessible, we will move the detailed Algorithm 1 to the Appendix and replace it with simplified pseudocode. To help visualize the workflow, we also created a diagram [Link](https://doi.org/10.6084/m9.figshare.31888510).  A user-friendly R package has already been included in the supplementary material.
>
> **4. Hyperparameter Sensitivity**: In the revised manuscript, we will explicitly distinguish between theoretically determined parameters and the single data-driven component $C$. We also supplement new simulation results [Link](https://doi.org/10.6084/m9.figshare.31888576), indicating that different hyperparameter choices in $\rho$ and using AIC and BIC for choosing $C$ yield almost identical results with differences at the 4th decimal place. We will revise the ``Hyperparameter Selection Strategy'' section with the following clarifications:
> * $\nu$ is governed by Theorems~4.1--4.2, which require $1/(2(q+1)) < \nu < 1/(2q)$ for optimal convergence. Under a standard smoothness assumption (e.g., $g_0$ is twice continuously differentiable, $q=2$), this restricts $\nu$ to $(0.167, 0.250)$. We therefore recommend $\nu = 0.2$ as a theoretically justified default, rather than treating it as a tunable parameter.
>  *  $\rho$: Theorem 4.1-4.2 only requires $\rho > 1$. The empirical results show negligible variation in performance across all streaming settings ($K$) for varying degrees of $\rho \in (1.5, 2.0, 2.5, 3.0)$. Therefore, we recommend $\rho = 2$ as a default.
>
> * $C$ (Scaling Constant): This is the only parameter that requires tuning. We determine it via an AIC-based procedure on the first data batch by search over $p_1 \in (3, \ldots, \lfloor n_1^{1/3} \rfloor)$ and let $C=p^*_1/n_1^{\nu}$. This provides a principled initialization and allows all subsequent updates to proceed automatically via $p_k = \lceil C  N_k^{\nu} \rceil$.
>
> **5.Privacy Guarantees**:  We agree that incorporating differential privacy (DP) is a valuable and nontrivial extension that warrants substantial future investigation. In the Discussion section, we will explicitly acknowledge that COLSA, in its current form, does not provide theoretical privacy guarantees despite sharing only summary statistics. Private online learning is a developing field. For example, both [3] for linear regression and [4] for more general statistical models introduce carefully designed perturbations to summary statistics to achieve formal privacy guarantees while maintaining statistical efficiency under some regularity conditions.
>
> These directions provide a promising foundation for extending COLSA with rigorous privacy protections and will be added in the future work section.
>
> [3] Online differentially private inference for linear regression model. Scand. J. Stat. 2026
>
> [4] Online differentially private inference in stochastic gradient descent. arXiv. 2025

---

> > ### Author Rebuttal · Reviewer_wqwf · 2026-04-03
> >
> > The authors have addressed my primary concerns in the rebuttal, and I have raised my score accordingly.

---

> > > ### Author Response · Authors · 2026-04-05
> > >
> > > Thank you for reviewing our rebuttal and for confirming that your concerns have been fully resolved. We truly appreciate your positive feedback and your stated intention to raise the score.
> > >
> > > It appears the system currently still reflects the original scores. We politely point this out in case you wish to update the system record. Thank you again for your valuable time and support

---

### Decision · Program_Chairs · 2026-04-30

**Decision:**

Accept (regular)

**Comment:**

This paper introduces a new online learning framework, Collaborative Operation of Linked Survival Analysis (COLSA), for Cox model, to solve the problem of non-decomposable likelihood functions. After rebuttal, all reviews are positive and main concerns have been addressed. There is a consensus that the paper should be accepted.